# E(3)-equivariant graph neural networks for data-efficient and accurate interatomic potentials

Simon Batzner [1✉], Albert Musaelian[1], Lixin Sun[1], Mario Geiger[2,3], Jonathan P. Mailoa[4], Mordechai Kornbluth [4], Nicola Molinari[1], Tess E. Smidt [5,6] & Boris Kozinsky [1,4✉]

This work presents Neural Equivariant Interatomic Potentials (NequIP), an E(3)-equivariant neural network approach for learning interatomic potentials from ab-initio calculations for molecular dynamics simulations. While most contemporary symmetry-aware models use invariant convolutions and only act on scalars, NequIP employs E(3)-equivariant convolutions for interactions of geometric tensors, resulting in a more information-rich and faithful representation of atomic environments. The method achieves state-of-the-art accuracy on a challenging and diverse set of molecules and materials while exhibiting remarkable data efficiency. NequIP outperforms existing models with up to three orders of magnitude fewer training data, challenging the widely held belief that deep neural networks require massive training sets. The high data efficiency of the method allows for the construction of accurate potentials using high-order quantum chemical level of theory as reference and enables high-fidelity molecular dynamics simulations over long time scales.

[1] John A. Paulson School of Engineering and Applied Sciences, Harvard University, Cambridge, MA 02138, USA. [2] École Polytechnique Fédérale de Lausanne, 1015 Lausanne, Switzerland. [3] Massachusetts Institute of Technology, Cambridge, MA 02139, USA. [4] Robert Bosch Research and Technology Center, Cambridge, MA 02139, USA. [5] Computational Research Division and Center for Advanced Mathematics for Energy Research Applications, Lawrence Berkeley National Laboratory, Berkeley, CA 94720, USA. [6] Department of Electrical Engineering and Computer Science and Research Laboratory of Electronics, Massachusetts Institute of Technology, Cambridge, MA 02139, USA. ✉email: batzner@g.harvard.edu; bkoz@seas.harvard

Molecular dynamics (MD) simulations are an indispensable tool for computational discovery in fields as diverse as energy storage, catalysis, and biological processes[1–3]. While the atomic forces required to integrate Newton's equations of motion can in principle be obtained with high fidelity from quantum-mechanical calculations such as density functional theory (DFT), in practice the unfavorable computational scaling of first-principles methods limits simulations to short time scales and small numbers of atoms. This prohibits the study of many interesting physical phenomena beyond the time and length scales that are currently accessible, even on the largest supercomputers. Owing to their simple functional form, classical models for the atomic potential energy can typically be evaluated orders of magnitude faster than first-principles methods, thereby enabling the study of large numbers of atoms over long time scales. However, due to their limited mathematical form, classical interatomic potentials, or force fields, are inherently limited in their predictive accuracy which has historically led to a fundamental trade-off between obtaining high computational efficiency while also predicting faithful dynamics of the system under study. The construction of flexible models of the interatomic potential energy based on machine learning, and in particular neural networks, has shown great promise in providing a way to move past this dilemma, promising to learn high-fidelity potentials from ab-initio reference calculations while retaining favorable computational efficiency[4–13]. Another central difference to classical force-fields based on analytical functions is that they often consist of explicit bonded and non-bonded terms, whereas machine learning interatomic potentials (ML-IPs) are agnostic to the bond topology of the system and treat all interactions in an identical manner, based on relative interatomic positions and the interacting chemical species. One of the limiting factors of neural network interatomic potentials (NN-IPs) is that they typically require large training sets of ab-initio calculations, often including thousands or even millions of reference structures[4,9,10,14–16]. This computationally expensive process of training data collection has severely limited the adoption of NN-IPs, as it quickly becomes a bottleneck in the development of force-fields for complex systems.

In this work, we present the Neural Equivariant Interatomic Potential (NequIP), a highly data-efficient deep learning approach for learning interatomic potentials from reference first-principles calculations. We show that the proposed method obtains high accuracy compared to existing ML-IP methods across a wide variety of systems, including small molecules, water in different phases, an amorphous solid, a reaction at a solid/gas interface, and a Lithium superionic conductor. Furthermore, we find that NequIP exhibits exceptional data efficiency, enabling the construction of accurate interatomic potentials from limited data sets of fewer than 1000 or even as little as 100 reference ab-initio calculations, where other methods require orders of magnitude more. It is worth noting that on small molecular data sets, NequIP outperforms not only other neural networks, but is also competitive with kernel-based approaches, which typically obtain better predictive accuracy than NN-IPs on small data sets (although at significant additional cost scaling in training and prediction). We further demonstrate high data efficiency and accuracy with state-of-the-art results on a training set of molecular data obtained at the quantum chemical coupled-cluster level of theory. Finally, we validate the method through a series of simulations and demonstrate that we can reproduce with high fidelity structural and kinetic properties computed from NequIP simulations in comparison to ab-initio molecular dynamics simulations (AIMD). We directly verify that the performance gains are connected with the unique E(3)-equivariant convolution architecture of the new NequIP model.

The first applications of machine learning for the development of interatomic potentials were built on descriptor-based approaches combined with shallow neural networks or Gaussian Processes[4,5], designed to exhibit invariance with respect to translation, permutation of atoms of the same chemical species, and rotation. Recently, rotationally invariant graph neural network interatomic potentials (GNN-IPs) have emerged as a powerful architecture for deep learning of interatomic potentials that eliminates the need for hand-crafted descriptors and allows to instead learn representations on graphs of atoms from invariant features of geometric data (e.g. radial distances or angles)[9–11,13]. In GNN-IPs, atomic structures are represented by collections of nodes and edges, where nodes in the graph correspond to individual atoms and edges are typically defined by simply connecting every atom to all other atoms that are closer than some cutoff distance $r_c$. Every node/atom $i$ is associated with a feature $\mathbf{h}_i \in \mathbb{R}^h$, consisting of scalar values, which is iteratively refined via a series of convolutions over neighboring atoms $j$ based on both the distance to neighboring atoms $r_{ij}$ and their features $\mathbf{h}_j$. This iterative process allows information to be propagated along the atomic graph through a series of convolutional layers and can be viewed as a message-passing scheme[17]. Operating only on interatomic distances allows GNN-IPs to be rotation- and translation-invariant, making both the output as well as features internal to the network invariant to rotations. In contrast, the method outlined in this work uses relative position vectors rather than simply distances (scalars) together with features comprised of not only scalars, but also higher-order geometric tensors. This makes internal features instead equivariant to rotation and allows for angular information to be used by rotationally equivariant filters. Similar to other methods, we can restrict convolutions to only a local subset of all other atoms that lie closer to the central atom than a chosen cutoff distance $r_c$, see Fig. 1, left.

A series of related methods have recently been proposed: DimeNet[11] expands on using pairwise interactions in a single convolution to include angular, three-body terms, but individual features are still comprised of scalars (distances and three-body angles are invariant to rotation), as opposed to vectors used in this work. Cormorant[18] uses an equivariant neural network for property prediction on small molecules. This method is demonstrated on potential energies of small molecules but not on atomic forces or systems with periodic boundary conditions. Townshend et al.[19] use the framework of Tensor-Field Networks[20] to directly predict atomic force vectors. The predicted forces are not guaranteed by construction to conserve energy since they are not obtained as gradients of the total potential energy. This may lead to problems in simulations of molecular dynamics over long times. None of these three works[11,18,19] demonstrates capability to perform molecular dynamics simulations. After a first version of this manuscript appeared online[21], a series of other equivariant GNN-IPs have been proposed, such as PaiNN[22] and NewtonNet[23]. Both of these methods were proposed after NequIP and only make use of $l = 1$ tensors. In addition, we also compare a series of other works that have since been proposed, including the GemNet[24], SpookyNet[25], and UNiTE approaches[26].

The contribution of the present work is the introduction of a deep learning energy-conserving interatomic potential for both molecules and materials built on E(3)-equivariant convolutions over geometric tensors that yields state-of-the-art accuracy, outstanding data-efficiency, and can with high fidelity reproduce structural and kinetic properties from molecular dynamics simulations.

## Results

**Equivariance**. The concept of equivariance arises naturally in machine learning of atomistic systems (see e.g.[27]): physical

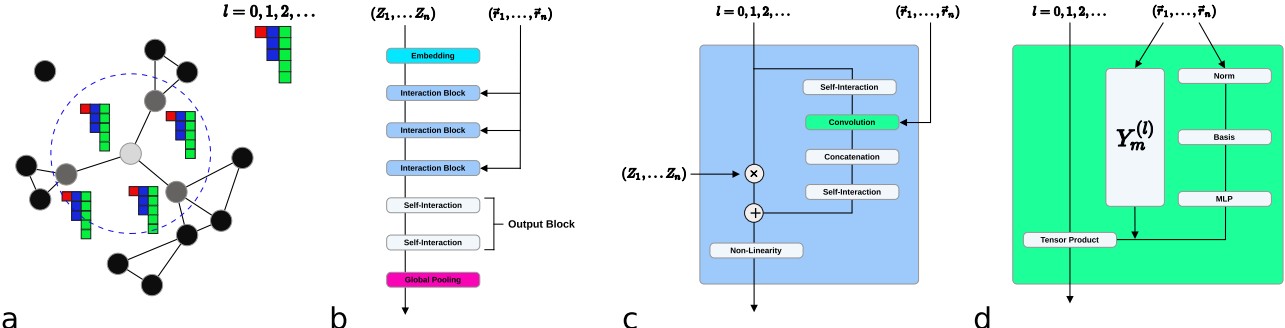

**Fig. 1 The NequIP network architecture.** From left to right: (**a**) a set of atoms is interpreted as an atomic graph with local neighborhoods (**b**) atomic numbers are embedded into $l = 0$ features, which are refined through a series of interaction blocks, creating scalar and higher-order tensor features. An output block then generates atomic energies, which are pooled to give the total predicted energy. **c** The interaction block, containing the convolution. **d** The convolution combines the product of the radial function $R(r)$ and the spherical harmonic projection of the unit vector $\hat{r}_{ij}$ with neighbouring features via a tensor product.

properties have well-defined transformation properties under translation, reflection, and rotation of a set of atoms. As a simple example, if a molecule is rotated in space, the vectors of its atomic dipoles or forces also rotate accordingly, via an equivariant transformation. Equivariant neural networks are able to more generally represent tensor properties and tensor operations of physical systems (e.g. vector addition, dot products, and cross products). Equivariant neural networks are guaranteed to preserve the known transformation properties of physical systems under a change of coordinates because they are explicitly constructed from equivariant operations. Formally, a function $f$: $X \rightarrow Y$ is equivariant with respect to a group $G$ that acts on $X$ and $Y$ if:

$$D_Y[g]f(x) = f(D_X[g]x) \quad \forall g \in G, \forall x \in X \quad (1)$$

where $D_X[g]$ and $D_Y[g]$ are the representations of the group element $g$ in the vector spaces $X$ and $Y$, respectively. Here, we focus on the effects of invariance and equivariance with respect to E(3), i.e. the group of rotations, reflections, and translations in 3D space.

**Neural equivariant interatomic potentials**. Given a set of atoms (a molecule or a material), we aim to find a mapping from atomic positions $\{\vec{r}_i\}$ and chemical species $\{Z_i\}$ to the total potential energy $E_{pot}$ and the forces acting on the atoms $\{\vec{F}_i\}$. Following previous work[4], this total potential energy is obtained as a sum of atomic potential energies. Forces are then obtained as the gradients of this predicted total potential energy with respect to the atomic positions (thereby guaranteeing energy conservation):

$$E_{pot} = \sum_{i \in N_{atoms}} E_{i,atomic} \quad (2)$$

$$\vec{F}_i = -\nabla_i E_{pot} \quad (3)$$

The atomic local energies $E_{i,atomic}$ are the scalar node attributes predicted by the graph neural network. Even though the output of NequIP is the predicted potential energy $E_{pot}$, which is invariant under translations, reflection, and rotations, the network contains internal features that are geometric tensors which are equivariant to rotation and reflection. This constitutes the core difference between NequIP and existing scalar-valued invariant GNN-IPs.

A series of methods has been introduced to realize rotationally equivariant neural networks[13,20,28–30]. Here, we build on the layers introduced in Tensor-Field Networks (TFN)[20], primitives for which are implemented in e3nn[31], which enable the construction of neural networks that exhibit invariance to translation and equivariance to parity, and rotation. Every atom

in NequIP is associated with features comprised of tensors of different orders: scalars, vectors, and higher-order tensors. Formally, the feature vectors are geometric objects that comprise a direct sum of irreducible representations of the O(3) symmetry group. The feature vectors $V_{acm}^{(l,p)}$ are indexed by keys $l, p$, where the "rotation order" $l = 0, 1, 2,...$ is a non-negative integer and parity is one of $p \in (1, -1)$ which together label the irreducible representations of O(3). The indices $a$, $c$, $m$, correspond to the atoms, the channels (elements of the feature vector), and the representation index which takes values $m \in [-l, l]$, respectively. The convolutions that operate on these geometric objects are equivariant functions instead of invariant ones, i.e. if a feature at layer $k$ is transformed under a rotation or parity transformation, then the output of the convolution from layer $k \rightarrow k + 1$ is transformed accordingly.

Convolution operations are naturally translation invariant, since their filters act on relative interatomic distance vectors. Moreover, they are permutation invariant since the sum over contributions from different atoms is invariant to permutations of those atoms. Note that while atomic features are equivariant to permutation of atom indices, globally, the total potential energy of the system is invariant to permutation. To achieve rotation equivariance, the convolution filters $S_m^{(l)}(\vec{r}_{ij})$ are constrained to be products of learnable radial functions and spherical harmonics, which are equivariant under SO(3)[20]:

$$S_m^{(l)}(\vec{r}_{ij}) = R(r_{ij}) Y_m^{(l)}(\hat{r}_{ij}) \quad (4)$$

where if $\vec{r}_{ij}$ denotes the relative position from central atom $i$ to neighboring atom $j$, $\hat{r}_{ij}$ and $r_{ij}$ are the associated unit vector and interatomic distance, respectively, and $S_m^{(l)}(\vec{r}_{ij})$ denotes the corresponding convolutional filter. It should be noted that all learnable weights in the filter lie in the rotationally invariant radial function $R(r_{ij})$. This radial function is implemented as a multi-layer perceptron which outputs together the radial weights for all filter-feature tensor production interactions:

$$R(r_{ij}) = W_n\sigma(...\sigma(W_2\sigma(W_1B(r_{ij})))) \quad (5)$$

where $B(r_{ij}) \in \mathbb{R}^{N_b}$ is a basis embedding of the interatomic distance of dimension $N_b$, $W_i$ are weight matrices and $\sigma(x)$ denotes the element-wise nonlinear activation function, for which we use the SiLU activation function[32] in our experiments. Radial Bessel functions and a polynomial envelope function $f_{env}$[11] are

used as the basis for the interatomic distances:

$$B(r_{ij}) = \frac{2}{r_c} \frac{\sin(\frac{b\pi}{r_c} r_{ij})}{r_{ij}} f_{env}(r_{ij}, r_c) \qquad (6)$$

where $r_c$ is a local cutoff radius, restricting interactions to atoms closer than some cutoff distance and $f_{env}$ is the polynomial defined in[11] with $p = 6$ operating on the interatomic distances normalized by the cutoff radius $\frac{r_{ij}}{r_c}$. The use of cutoffs/local atomic environments allows the computational cost of evaluation to scale linearly with the number of atoms. Similar to[11], at network initialization, the Bessel roots are set as $b = [1, 2, ..., N_b]$, where $N_b$ is the number of basis functions, and we subsequently optimize $b\pi$ via backpropagation rather than keeping it constant. For systems with periodic boundary conditions, we use neighbor lists as implemented in the ASE code[33] to identify appropriate atomic neighbors.

Finally, in the convolution, the input atomic feature tensor and the filter have to again be combined in an equivariant manner, which is achieved via a geometric tensor product that yields an output feature that again is rotationally equivariant. A tensor product of two geometric tensors is computed via contraction with the Clebsch-Gordan coefficients, as outlined in[20]. A tensor product between an input feature of order $l_i$ and a convolutional filter of order $l_f$ yields irreducible representations of output orders $|l_i - l_f| \le l_o \le |l_i + l_f|$. In NequIP, we use a maximum rotation order $l_{max}$ and discard all tensor product operations that would results in irreducibe representations with $l_o > l_{max}$. Omitting all higher-order interactions that go beyond the $0 \otimes 0 \to 0$ interaction will result in a conventional GNN-IP with invariant convolutions over scalar features, similar to e.g. SchNet[9].

The final symmetry the network needs to respect is that of parity: how the tensor transforms under inversion, i.e. $\vec{x} \to -\vec{x}$. A tensor has even parity ($p = 1$) if it is invariant to such a transformation; it has odd parity ($p = -1$) if its sign flips under that transformation. Parity equivariance is achieved by only allowing contributions from a filter and an incoming tensor feature with parities $p_f$ and $p_i$ to contribute to an output feature if the following selection rule is satisfied:

$$p_o = p_i p_f \qquad (7)$$

Finally, as outlined in[20], a full convolutional layer $\mathcal{L}$ implementing an interaction with filter $f$ acting on an input $i$ producing output $o$: $l_i \otimes l_f \to l_o$ is given by:

$$\mathcal{L}^{l_o, p_o, l_f, p_f, l_i, p_i}_{acm_o}\left(\vec{r}_a, V^{l_i, p_i}_{acm_i}\right) = \sum_{m_f, m_i} C^{l_o, m_o}_{l_i, m_i, l_f, m_f} \sum_{b \in S} (R(r_{ab})_{c, l_o, p_o, l_f, p_f, l_i, p_i}$$
$$Y^{l_f}_{m_f}(\hat{r}_{ab}) V^{l_i, p_i}_{bcm_i} \qquad (8)$$

where $a$ and $b$ index the central atom of the convolution and the neighboring atom $b \in S$, respectively, and $C$ indicates the Clebsch-Gordan coefficients. It should be noted that the placement of indices into sub- and superscript does not carry specific meaning. Note that the Clebsch-Gordan coefficients do not depend on the parity of the arguments. There can be multiple $\mathcal{L}^{l_o, p_o}_{acm_o}$ tensors for a given output rotation order and parity ($l_o, p_o$) resulting from different combinations of ($l_i, p_i$) and ($l_f, p_f$); we take all such possible output tensors with $l_o \le l_{max}$ and concatenate them. We also divide the output of the sum over neighbors by $\sqrt{N}$, where $N$ denotes the average number of neighbors of an atom. To update the atomic features, the model also uses dense layers that are applied in an atom-wise fashion with weights shared across atoms, similar to the self-interaction layers in SchNet[9]. While different weights are used for different rotation

orders, the same set of weights is applied for all representation indices $m$ of a given tensor with rotation order $l$ to maintain equivariance.

The NequIP network architecture, shown in Fig. 1, is built on an atomic embedding, followed by a series of interaction blocks, and finally an output block:

- Embedding: following SchNet, the initial feature is generated using a trainable embedding that operates on the atomic number $Z_i$ (represented via a one-hot encoding) alone, implemented via a trainable self-interaction layer.
- Interaction Block: interaction blocks encode interactions between neighboring atoms: the core of this block is the convolution function, outlined in equation (8). Features from different tensor product interactions that yield the same rotation and parity pair ($l_o, p_o$) are mixed by linear atom-wise self-interaction layers. We equip interaction blocks with a ResNet-style update[34]: $\mathbf{x^{k+1}} = f(\mathbf{x^k}) + \text{Self - Interaction}(\mathbf{x^k})$, where $f$ is the series of self-interaction, convolution, concatenation, and self-interaction. The weights of the Self - Interaction in the preceding formula are learned separately for each species. Finally, the mixed features are processed by an equivariant SiLU-based gate nonlinearity[28,32] (even and odd scalars are not gated, but instead are processed directly by SiLU and tanh nonlinearities, respectively).
- Output Block: the $l = 0$ features of the final convolution are passed to an output block, which consists of a set of two atom-wise self-interaction layers.

For each atom the final layer outputs a single scalar, which is interpreted as the atomic potential energy. These are then summed to give the total predicted potential energy of the system (Equation (2)). Forces are subsequently obtained as the negative gradient of the predicted total potential energy, thereby ensuring both energy conservation and rotation-equivariant forces (see equation (3)).

**Experiments**. We validate the proposed method on a diverse series of challenging data sets: first we demonstrate that we improve upon state-of-the-art accuracy on MD-17, a data set of small, organic molecules that is widely used for benchmarking ML-IPs[9,11,35–37]. Next, we show that NequIP can accurately learn forces obtained on small molecules at the quantum chemical CCSD(T) level of theory[37]. To broaden the applicability of the method beyond small isolated molecules, we finally explore a series of extended systems with periodic boundary conditions, consisting of both surfaces and bulk materials: water in different phases[15,38], a chemical reaction at a solid/gas interface, an amorphous Lithium Phosphate[12], and a Lithium superionic conductor[13]. Details of the training procedure are provided in the Methods section.

**MD-17 small molecule dynamics**. We first evaluate NequIP on MD-17[35–37], a data set of small organic molecules in which reference values of energy and forces are generated by ab-initio MD simulations with DFT. Recently, a recomputed version of the original MD-17 data with higher numerical accuracy has been released, termed the revised MD-17 data set[39] (an example histogram of potential energies and force components can be found in the Supplementary Information). In order to be able to compare results to a wide variety of methods, we benchmark NequIP on both data sets. For training and validation, we use a combined N=1,000 configurations. The mean absolute error in the energies and force components is shown in Tables 1 and 2. We compare results using NequIP with those from published leading MLIP

**Table 1 Energy and Force MAE for molecules on the original MD-17 data set, reported in units of [meV] and [meV/Å], respectively, and a training budget of 1000 reference configurations.**

| Molecule | | SchNet | DimeNet | sGDML | PaiNN | SpookyNet | GemNet-(T/Q) | NewtonNet | UNiTE | NequIP ($l = 3$) |
|---|---|---|---|---|---|---|---|---|---|---|
| Aspirin | Energy | 16.0 | 8.8 | 8.2 | 6.9 | 6.5 | – | 7.3 | – | **5.7** |
| | Forces | 58.5 | 21.6 | 29.5 | 14.7 | 11.2 | 9.4 | 15.1 | **6.8** | 8.0 |
| Ethanol | Energy | 3.5 | 2.8 | 3.0 | 2.7 | 2.3 | – | 2.6 | – | **2.2** |
| | Forces | 16.9 | 10.0 | 14.3 | 9.7 | 4.1 | 3.7 | 9.1 | 4.0 | **3.1** |
| Malonaldehyde | Energy | 5.6 | 4.5 | 4.3 | 3.9 | 3.4 | – | 4.2 | – | **3.3** |
| | Forces | 28.6 | 16.6 | 17.8 | 13.8 | 7.2 | 6.7 | 14.0 | 6.9 | **5.6** |
| Naphthalene | Energy | 6.9 | 5.3 | 5.2 | 5.0 | 5.0 | – | 5.1 | – | **4.9** |
| | Forces | 25.2 | 9.3 | 4.8 | 3.3 | 3.9 | 2.2 | 3.6 | 2.8 | **1.7** |
| Salicylic acid | Energy | 8.7 | 5.8 | 5.2 | 4.9 | 4.9 | – | 5.0 | – | **4.6** |
| | Forces | 36.9 | 16.2 | 12.1 | 8.5 | 7.8 | 5.4 | 8.5 | 4.2 | **3.9** |
| Toluene | Energy | 5.2 | 4.4 | 4.3 | 4.1 | 4.1 | – | 4.1 | – | **4.0** |
| | Forces | 24.7 | 9.4 | 6.1 | 4.1 | 3.8 | 2.6 | 3.8 | 3.1 | **2.0** |
| Uracil | Energy | 6.1 | 5.0 | 4.8 | **4.5** | 4.6 | – | 4.6 | – | **4.5** |
| | Forces | 24.3 | 13.1 | 10.4 | 6.0 | 5.2 | 4.2 | 6.5 | 4.2 | **3.3** |

For GemNet, the best result out of the T/Q versions is presented and for PaiNN the best between force-only and joint force and energy training. For UNiTE, we compare to the "direct-learning" results reported in[26].
Best results are marked in bold.

**Table 2 Energy and Force MAE for molecules on the revised MD-17 data set, reported in units of [meV] and [meV/Å], respectively, and a training budget of 1000 reference configurations.**

| Molecule | | FCHL19 | UNiTE | GAP | ANI | ACE | GemNet-(T/Q) | NequIP ($l = 0$) | NequIP ($l = 1$) | NequIP ($l = 2$) | NequIP ($l = 3$) |
|---|---|---|---|---|---|---|---|---|---|---|---|
| Aspirin | Energy | 6.2 | 2.4 | 17.7 | 16.6 | 6.1 | – | 25.2 | 3.8 | 2.4 | **2.3** |
| | Forces | 20.9 | **7.6** | 44.9 | 40.6 | 17.9 | 9.5 | 42.2 | 12.6 | 8.5 | 8.2 |
| Azobenzene | Energy | 2.8 | 1.1 | 8.5 | 15.9 | 3.6 | – | 20.3 | 1.1 | 0.8 | **0.7** |
| | Forces | 10.8 | 4.2 | 24.5 | 35.4 | 10.9 | – | 34.4 | 4.5 | 3.3 | **2.9** |
| Benzene | Energy | 0.3 | 0.07 | 0.75 | 3.3 | **0.04** | – | 3.2 | 0.09 | 0.06 | **0.04** |
| | Forces | 2.6 | 0.73 | 6.0 | 10.0 | 0.5 | 0.5 | 10.3 | 0.4 | 0.4 | **0.3** |
| Ethanol | Energy | 0.9 | 0.62 | 3.5 | 2.5 | 1.2 | – | 2.0 | 1.0 | 0.5 | **0.4** |
| | Forces | 6.2 | 3.7 | 18.1 | 13.4 | 7.3 | 3.6 | 11.9 | 6.5 | 3.5 | **2.8** |
| Malonaldehyde | Energy | 1.5 | 1.1 | 4.8 | 4.6 | 1.7 | – | 4.4 | 1.6 | 0.9 | **0.8** |
| | Forces | 10.2 | 6.6 | 26.4 | 24.5 | 11.1 | 6.6 | 23.2 | 10.3 | 5.9 | **5.1** |
| Naphthalene | Energy | 1.2 | 0.46 | 3.8 | 11.3 | 0.9 | – | 14.7 | 0.4 | 0.3 | **0.2** |
| | Forces | 6.5 | 2.6 | 16.5 | 29.2 | 5.1 | 1.9 | 20.6 | 2.1 | 1.4 | **1.3** |
| Paracetamol | Energy | 2.9 | 1.9 | 8.5 | 11.5 | 4.0 | – | 17.5 | 2.1 | **1.4** | **1.4** |
| | Forces | 12.2 | 7.1 | 28.9 | 30.4 | 12.7 | – | 33.6 | 9.3 | **5.9** | **5.9** |
| Salicylic acid | Energy | 1.8 | 0.73 | 5.6 | 9.2 | 1.8 | – | 11.4 | 1.0 | 0.8 | **0.7** |
| | Forces | 9.5 | 3.8 | 24.7 | 29.7 | 9.3 | 5.3 | 29.8 | 5.7 | 4.2 | **4.0** |
| Toluene | Energy | 1.6 | 0.45 | 4.0 | 7.7 | 1.1 | – | 9.7 | 0.5 | **0.3** | **0.3** |
| | Forces | 8.8 | 2.5 | 17.8 | 24.3 | 6.5 | 2.2 | 26.6 | 2.6 | 1.8 | **1.6** |
| Uracil | Energy | 0.6 | 0.58 | 3.0 | 5.1 | 1.1 | – | 10.0 | 0.6 | **0.4** | **0.4** |
| | Forces | 4.2 | 3.8 | 17.6 | 21.4 | 6.6 | 3.8 | 26.0 | 4.1 | **2.9** | 3.1 |

For GemNet, the best result out of the T/Q versions is presented. For FCHL19, the best results between energy-only, force-only and joint force and energy training are presented. For UNiTE, we compare to the "direct-learning" results reported in[26].
Best results are marked in bold.

models. We find that NequIP significantly outperforms invariant GNN-IPs (such as SchNet[9] and DimeNet[11]), shallow neural networks (such as ANI[40]), and kernel-based approaches (such as GAP[5], FCHL19/GPR[39,41] and sGDML[37]). Finally, we compare to a series of other methods including ACE[42], SpookyNet[25], and GemNet[24] as well as other equivariant neural networks such as PaiNN[22], NewtonNet[23], and UNiTE[26]. Again, it should be stressed that PaiNN and NewtonNet are $l_{max} = 1$-only versions of equivariant networks. The results for ACE, GAP, and ANI on the revised MD-17 data set are those reported in[43]. Importantly, we train and test separate NequIP models on both the original and the revised MD-17 data set, and find that NequIP obtains significantly lower energy errors on the revised data set, while the force accuracy is similar on the two data sets. In line with

previous work[39], this suggests that the noise floor on the original MD-17 data is higher on the energies and that only the results on the revised MD-17 data set should be used for comparing different methods.

Remarkably, we find that NequIP outperforms all other methods. The consistent improvements in accuracy compared to sGDML and FCHL19/GPR are particularly surprising, as these are based on kernel methods, which typically obtain better performance than deep neural networks on small training sets. We run a convergence scan on the rotation order $l \in \{0, 1, 2, 3\}$ and find that increasing the tensor rank beyond $l = 1$ gives a consistent improvement. The significant improvement from $l = 0$ to $l = 1$ highlights the crucial role of equivariance in obtaining improved accuracy on this task.

**Table 3 RMSE of energies and forces on liquid water and the three ices in units of [meV/molecule] and [meV/Å], with energy errors normalized by the number of molecules in the system.**

| System | | NequIP, a) | NequIP, b) | NequIP, c) | DeepMD |
|---|---|---|---|---|---|
| Liquid Water | Energy | – | 1.6 | 1.7 | 1.0 |
| | Forces | 11.9 | 49.4 | 11.6 | 40.4 |
| Ice Ih (b) | Energy | – | 2.5 | 4.3 | 0.7 |
| | Forces | 10.2 | 55.8 | 9.9 | 43.3 |
| Ice Ih (c) | Energy | – | 3.9 | 10.2 | 0.7 |
| | Forces | 12.0 | 27.7 | 11.7 | 26.8 |
| Ice Ih (d) | Energy | – | 2.6 | 12.7 | 0.8 |
| | Forces | 9.8 | 23.2 | 9.5 | 25.4 |

Note that the NequIP models were trained on <0.1% of the training data of DeepMD. NequIP model (a) refers to loss function weighting $\lambda_F = 1$, $\lambda_E = 0$, model (b) to $\lambda_F = 100$, $\lambda_E = 1$, and model c) to $\lambda_F = 100,000$, $\lambda_E = 1$.

**Force training at quantum chemical accuracy**. The ability to achieve high accuracy on a comparatively small data set facilitates easier development of Machine Learning Interatomic Potentials on expensive high-order ab-initio quantum chemical methods, such as e.g. the coupled cluster method CCSD(T). However, the high computational cost of CCSD(T) has thus far hindered the use of reference data structures at this level of theory, prohibited by the need for large data sets that are required by available NN-IPs. Leveraging the high data efficiency of NequIP, we evaluate it on a set of molecules computed at quantum chemical accuracy (aspirin at CCSD, all others at CCSD(T))[37] and compare the results to those reported for sGDML[37] and GemNet[24]. Results are show in the Supplementary Information.

**Liquid water and ice dynamics**. To demonstrate the applicability of NequIP beyond small molecules, we evaluate the method on a series of extended systems with periodic boundary conditions. As a first example we use a joint data set consisting of liquid water and three ice systems[15,38] computed at the PBE0-TS level of theory. This data set[15] contains: (a) liquid water, $P = 1$ bar, $T = 300$ K, computed via path-integral AIMD, (b) ice Ih, $P = 1$ bar, $T = 273$ K, computed via path-integral AIMD (c) ice Ih, $P = 1$ bar, $T = 330$ K, computed via classical AIMD (d) ice Ih, $P = 2.13$ kbar, $T = 238$ K, computed via classical AIMD. A DeepMD NN-IP model was previously trained[15] for water and ice using a joint training set containing 133,500 reference calculations of these four systems. To assess data efficiency of the NequIP architecture, we similarly train a model jointly on all four parts of the data set, but using only 133 structures for training, i.e. 1000x fewer data. The 133 structures were sampled randomly following a uniform distribution from the full data set available online which consists of water and ice structures and is made up of a total of 140,000 frames, coming from the same MD trajectories that were used in the earlier work[15]. Table 3 compares the energy and force errors of NequIP trained on the 133 structures vs DeepMD trained on 133,500 structures. We find that with 1000x fewer training data NequIP significantly outperforms DeepMD on all four parts of the data set in the error on the force components. We note that there are $3N$ force components for each training frame but only one energy target. Consequently, one would except that on energies the much larger training set used for DeepMD would results in an even stronger difference. We find that while this is indeed the case, the NequIP results on the liquid phase are surprisingly competitive. Finally, we report results using three different weightings of energies and forces in the loss function and see that increasing the energy weighting results in significantly improved energy errors at the cost of a small increase in force error. We note that the version of DeepMD published in[15] is not smooth, and a smooth version has

since been proposed[44]. However,[44] does not report results on the water/ice systems. It would be of interest to investigate the performance of the smooth DeepMD version as a function of training set size.

**Heterogeneous catalysis of formate dehydrogenation**. Next, we apply NequIP to a catalytic surface reaction. In particular, we investigate the dynamics of formate undergoing dehydrogenation decomposition $(HCOO^* \rightarrow H^* + CO_2)$ on a Cu < 110 > surface (see Fig. 2). This system is highly heterogeneous: it has both metallic and covalent types of bonding as well as charge transfer between the metal and the molecule, making it a particularly challenging test system. Different states of the molecule also lead to dissimilar C-O bond lengths[45,46]. Training structures consist of 48 Cu atoms and 4 atoms of the molecule (HCOO* or CO$_2$+H*). A NequIP model trained on 2,500 structures obtains MAEs in the force components of 19.9 meV/Å, 71.3 meV/Å, 13.0 meV/Å, and 47.6 meV/Å, on the four elements C, O, H, and Cu, respectively. We find from this an average force MAE of 38.4 meV/Å, equally weighted over these four per-species MAEs, as well as an energy MAE of 0.50 meV/atom, demonstrating that NequIP is able to accurately model the interatomic forces for this complex reactive system. A more detailed analysis of the resulting dynamics will be the subject of a separate study.

**Lithium phosphate amorphous glass formation**. To examine the ability of the model to capture dynamical properties, we demonstrate that NequIP can describe structural dynamics in amorphous lithium phosphate with composition $Li_4P_2O_7$. This material is a member of the promising family of solid electrolytes for Li-metal batteries[12,47,48], with non-trivial Li-ion transport and phase transformation behaviors. The data set consists of two 50 ps long AIMD simulations: one of the molten structure at T = 3000 K and another of a quenched glass structure at $T = 600$ K. We train NequIP on a subset of 1000 structures from the molten trajectory. Table 4 shows the error in the force components on both the test set from the AIMD molten trajectory and the full AIMD quenched glass trajectory. To then evaluate the physical fidelity of the trained model, we use it to run a set of ten MD simulations of length 50 ps at $T = 600$ K in the NVT ensemble and compare the total radial distribution function (RDF) without element distinction as well as the angular distribution functions (ADF) of the P–O–O (P central atom) and O–P–P (O central atom) angles averaged over ten runs to the ab-inito trajectory at the same temperature. The P–O–O angle corresponds to the tetrahedral bond angle, while the O–P–P corresponds to a bridging angle between corner-sharing phosphate tetrahedra (Fig. 2). Fig. 3 shows that NequIP can accurately reproduce the RDF and the two ADFs, in comparison with AIMD, after training on only 1000 structures.

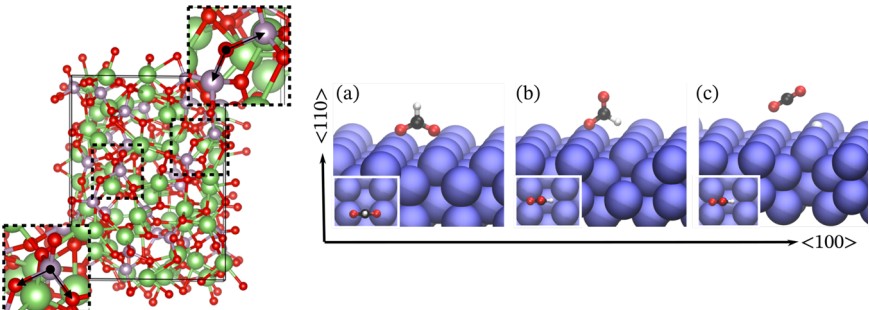

**Fig. 2 Benchmark systems.** Left: Quenched glass structure of $Li_4P_2O_7$, including the tetrahedral bond angle (bottom left) and the bridging angle between corner-sharing phosphate tetrahedra (top right). Right: The formate on Cu system. Perspective view of atomic configurations of (**a**) bidentate HCOO (**b**) monodentate HCOO and (**c**) $CO_2$ and a hydrogen adatom on a Cu(110) surface. The blue, red, black, and white spheres represent Cu, O, C, and H atoms, respectively. The subset shown in each subplot is the corresponding top view along the <110> orientation.

**Table 4 NequIP E/F MAE/RMSE for LiPS and $Li_4P_2O_7$ for different data set sizes in units of [meV/Å] and [meV/atom].**

| System | Data set size | | MAE | RMSE |
|---|---|---|---|---|
| LiPS | 10 | Energy | 2.03 | 2.54 |
| | | Forces | 97.8 | 132.4 |
| LiPS | 100 | Energy | 0.44 | 0.56 |
| | | Forces | 25.8 | 35.0 |
| LiPS | 1000 | Energy | 0.12 | 0.15 |
| | | Forces | 7.7 | 10.8 |
| LiPS | 2500 | Energy | 0.08 | 0.10 |
| | | Forces | 4.7 | 6.5 |
| $Li_4P_2O_7$, melt | 1000 | Energy | 0.4 | 0.8 |
| | | Forces | 34.0 | 59.5 |
| $Li_4P_2O_7$, quench | 1000 | Energy | 0.5 | 0.5 |
| | | Forces | 21.3 | 34.9 |

The model for $Li_4P_2O_7$ was trained exclusively on structures from the melted trajectory. The reported test errors for the melt are computed on the remaining set of structures from the full melt trajectory; errors for the quench are computed on the full quench trajectory.

This demonstrates that the model generates the glass state and recovers its dynamics and structure almost perfectly, despite having seen only the high-temperature molten training data. We also include results from a longer NequIP-driven MD simulation of 500 ps, which can be found in the SI.

**Lithium thiophosphate superionic transport.** To show that NequIP can model kinetic transport properties from small training sets at high accuracy, we study Li-ion diffusivity in LiPS ($Li_{6.75}P_3S_{11}$), a crystalline superionic Li conductor consisting of a simulation cell of 83 atoms[13]. MD is widely used to study diffusion; training a ML-IP to the accuracy required to predict kinetic properties, however, has in the past required large training set sizes ([49] e.g. uses a data set of 30,874 structures to study Li diffusion in $Li_3PO_4$). Here we demonstrate that not only does NequIP obtain small errors in the energies and force components, but it also accurately predicts the diffusivity after training on a data set obtained from an AIMD simulation. Again, we find that very small training sets lead to highly accurate models, as shown in Table 4 for training set sizes of 10, 100, 1000 and 2500 structures. We run a series of MD simulations with the NequIP potential trained on 2500 structures in the NVT ensemble at the same temperature as the AIMD simulation for a total simulation time of 50 ps and a time step of 0.25 fs, which we found advantageous for the reliability and stability of long simulations. We measure the Li diffusivity in these NequIP-driven MD simulations (computed via the slope of the mean square

displacement) started from different initial velocities, randomly sampled from a Maxwell-Boltzmann distribution. We find a mean diffusivity of $1.25 \times 10^{-5} cm^2/s$, in excellent agreement with the diffusivity of $1.37 \times 10^{-5} cm^2/s$ computed from AIMD, thus achieving a relative error of as little as 9%. Fig. 4 shows the mean square displacements of Li for an example run of NequIP in comparison to AIMD.

**Data efficiency.** In the above experiments, NequIP exhibits exceptionally high data efficiency. It is interesting to consider the reasons for such high performance and verify that it is connected to the equivariant nature of the model. First, it is important to note that each training configuration contains multiple labels: in particular, for a training set of $M$ first-principles calculations with structures consisting of $N$ atoms, the energies and force components together give a total of $M(3N + 1)$ labels. In order to gain insight into the reasons behind increased accuracy and data efficiency, we perform a series of experiments with the goal of isolating the effect of using equivariant convolutions. In particular, we run a set of experiments in which we explicitly turn on or off interactions of higher order than $l = 0$. This defines two settings: first, we train the network with the full set of tensor features up to a given order $l$ and the corresponding equivariant interactions. Second, we turn off all interactions involving $l > 0$, making the network a conventional invariant GNN-IP, involving only invariant convolutions over scalar features in a SchNet-style fashion.

As a first test system we choose bulk water: in particular we use the data set introduced in[50]. We train a series of networks with identical hyperparameters, but vary the training set sizes between 10 and 1000 structures. As shown in Fig. 5, we find that the equivariant networks with $l \in 1, 2, 3$ significantly outperform the invariant networks with $l = 0$ for all data set sizes as measured by the MAE of force components. This suggests that it is indeed the use of tensor features and equivariant convolutions that enables the high sample efficiency of NequIP. In addition, it is apparent that the learning curves of equivariant networks have a different slope in log-log space. It has been observed that learning curves typically follow a power-law of the form[51]: $\epsilon \propto aN^b$ where $\epsilon$ and $N$ refer to the generalization error and the number of training points, respectively. The exponent of this power-law (or equivalently the slope in log-log space) determines how fast a learning algorithm learns as new data become available. Empirical results have shown that this exponent typically remains fixed across different learning algorithms for a given data set, and different methods only *shift* the learning curve, leaving the log-log slope unaffected[51]. The same trend can also be observed for various methods on the aspirin molecule in the MD-17 data set

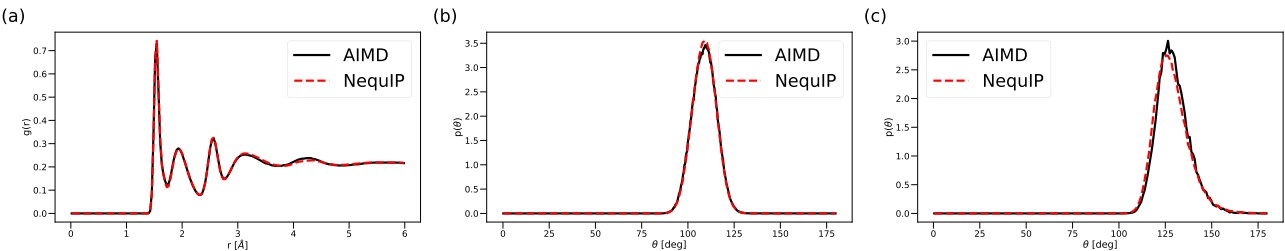

**Fig. 3 Structure of Li₄P₂O₇. a** Radial Distribution Function, (**b**) Angular Distribution Function, tetrahedral bond angle, (**c**) Angular Distribution Function, bridging oxygen. All are defined as probability density functions; NequIP results are averaged over 10 runs with different initial velocities.

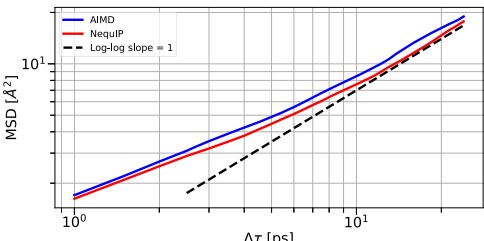

**Fig. 4 Lithium Kinetics.** Comparison of the Li MSD of AIMD and an example NequIP trajectory of $Li_{6.75}P_3S_{11}$.

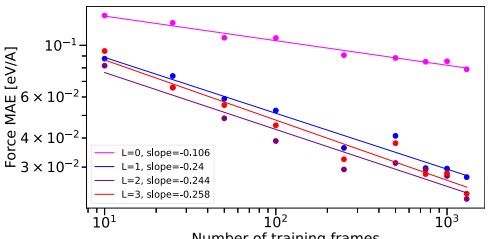

**Fig. 5 Learning curves.** Log-log plot of the predictive error on the water data set from[50] using NequIP with $l \in \{0, 1, 2, 3\}$ as a function of training set size, measured via the force MAE.

(see Supplementary Information) where across a series of descriptors and regression models (sGDML, FCHL19, and PhysNet[10,37,41]) the learning curves show an approximately similar log-log slope (results obtained from http://quantum-machine.org/gdml/#datasets). To our surprise, we observe that the equivariant NequIP networks break this pattern. Instead they follow a log-log slope with larger magnitude, meaning that they learn faster as new data become available. An invariant $l = 0$ NequIP network, however, displays a similar log-log slope to other methods, suggesting that it is indeed the equivariant nature of NequIP that allows for the change in learning behavior. Further increasing the rotation order $l$ beyond $l = 1$ again only shifts the learning curve and does not results in an additional change in log-log slope. To control for the different number of weights and features of networks of different rotation order $l$, we report weight- and feature-controlled data in the SI. Both show qualitatively the same effect. The SI also contains results on the behavior of the energies, when trained jointly with forces. For details on the training setup and the control experiments, see the Methods section.

We further note, that in[50], a Behler-Parrinello Neural Network (BPNN) was trained on 1303 structures, yielding a RMSE of ≈120 meV/Å in forces when evaluated on the remaining 290 structures. We find that NequIP $l = 2$ models trained with as little as 100 and 250 data points obtain RMSEs of 123.3 meV/Å and 98.3 meV/Å respectively (note that Fig. 5 shows the MAE). This provides further evidence that NequIP exhibits significantly improved data efficiency in comparison with existing methods.

## Discussion

This work introduces NequIP, a novel Machine Learning method for computing the potential energy and atomic forces of molecules and materials based on E(3)-Equivariant Neural Networks. The findings lead to a series of interesting questions to consider: of particular interest is the sample efficiency of the equivariant NequIP network when compared to the more widely used invariant representations. In addition to questions around the

effect of equivariance on accuracy and learning dynamics, a clear theoretical understanding of how the many-body character of interactions arises in message passing interatomic potentials remains elusive. Further, a promising direction for future work is to investigate the potential benefits of explicitly including long-range interactions and to measure to what extent - if any - these might be captured by the message passing mechanism. Finally, while we find that NequIP displays excellent predictive accuracy, generalization to unseen phases, and remarkably high sample efficiency, an open challenge that remains is the interpretability of deep learning interatomic potentials. Energy contributions in classical interatomic potentials can be explicitly assigned to individual types of interactions, such as pair-wise bonded terms or Coulomb or van der Waals non-bonded interactions. The potential benefits and optimal ways of including such physical knowledge explicitly into the complex functional forms underlying deep learning interatomic potentials still need to be systematically explored. On the other hand, the simplicity of the functional form of classical force-fields that allows for this level of interpretability severely limits their accuracy, presenting an interesting tension between the two approaches. We expect the proposed method will enable researchers in computational chemistry, physics, biology, and materials science to conduct molecular dynamics simulations of complex reactions and phase transformations at increased accuracy and efficiency.

## Methods

Software: All experiments were run with the `nequip` software available at github.com/mir-group/nequip in version 0.3.3, git commit `50ddbfc31bd44e267b7b b7d2d36d76417b0885ec`. In addition, the `e3nn` library[31] was used under version 0.3.5, PyTorch under version `1.9.0`[52], PyTorch Geometric under version 1.7.2[53], and Python under version 3.9.6.

Reference Data Sets:

original MD-17: MD-17[35–37] is a data set of eight small organic molecules, obtained from MD simulations at $T = 500K$ and computed at the PBE+vdW− TS level of electronic structure theory, resulting in data set sizes between 133,770 and 993,237 structures. The data set was obtained from http://quantum-machine.org/gdml/#datasets. For each molecule, we use 950 configurations for training and 50 for validation, sampled uniformly from the full data set, and evaluate the test error on all remaining configurations in the data set.

**Table 5 Tensor rank *l*, feature size, radial cutoff in units of [Å], as well as energy and force weights used in the joint loss function.**

| Data Set | Tensor rank *l* | # Features | $r_c$ | $\lambda_E$ | $\lambda_F$ |
|---|---|---|---|---|---|
| MD-17 | 3 | 64 | 4.0 | 1 | 1000 |
| revMD-17 | {0, 1, 2, 3} | 64 | 4.0 | 1 | 1000 |
| CCSD/CCSD(T) | 3 | 64 | 4.0 | 1 | 1000 |
| Water +Ices, DeepMD | 2 | 32 | 6.0 | see 3 | see 3 |
| Formate on Cu | 2 | 32 | 5.0 | 1 | 2704 |
| $Li_4P_2O_7$ | 2 | 32 | 5.0. | 1 | 43,264 |
| LiPS | 2 | 32 | 5.0 | 1 | 6889 |
| Water, Cheng et al. | {0, 1, 2, 3} | 32 | 4.5 | 1 | 36,864 |

All models were trained with even and odd features, i.e. a tensor rank of $l = 1$ and 32 features corresponds to $32 \times 0o + 32 \times 0e + 32 \times 1o + 32 \times 1e$. The force weightings for formate on Cu, LiPO, LiPS, and the water system for sample efficiency tests stem from $N_{atoms}^2$.

revised MD-17: The revised MD-17 data set is a recomputed version of MD-17 obtained at the PBE/def2-SVP level of theory. Using a very tight SCF convergence as well as a very dense DFT integration grid, 100,000 structures[39] of the original MD-17 data set were recomputed. The data set can be downloaded at https://figshare.com/articles/dataset/Revised_MD17_dataset_rMD17_/12672038. For each molecule, we use 950 configurations for training and 50 for validation, sampled uniformly from the full data set, and evaluate the test error on all remaining configurations in the data set.

Molecules@CCSD/CCSD(T): The data set of small molecules at CCSD and CCSD(T) accuracy[37] contains positions, energies, and forces for five different small molecules: Asprin (CCSD), Benzene, Malonaldehyde, Toluene, Ethanol (all CCSD(T)). Each data set consists of 1500 structures with the exception of Ethanol, for which 2000 structures are available. For more detailed information, we direct the reader to[37]. The data set was obtained from http://quantum-machine.org/gdml/#datasets. The training/validation set consists of a total of 1000 molecular structures which we split into 950 for training and 50 for validation (sampled uniformly), and we test the accuracy on all remaining structures (we use the train/test split provided with the data set, but further split the training set into training and validation sets).

Liquid Water and Ice: The data set of liquid waters and ice structures[15,38] was generated from classical AIMD and path-integral AIMD simulations at different temperatures and pressures, computed with a PBE0-TS functional[15]. The data set contains a total of 140,000 structures, of which 100,000 are liquid water and 20,000 are Ice Ih b),10,000 are Ice Ih c), and another 10,000 are Ice Ih d). The liquid water system consists of 64 $H_2O$ molecules (192 atoms), while the ice structures consist of 96 $H_2O$ molecules (288 atoms). We use a validation set of 50 frames and report the test accuracy on all remaining structures in the data set.

Formate decomposition on Cu: The decomposition process of formate on Cu involves configurations corresponding to the cleavage of the C-H bond, initial and intermediate states (monodentate, bidentate formate on Cu <110>) and final states (H ad-atom with a desorbed $CO_2$ in the gas phase). Nudged elastic band (NEB) method was first used to generate an initial reaction path of the C-H bond breaking. 12 short ab initio molecular dynamics, starting from different NEB images, were run to collect a total of 6855 DFT structures. The CP2K[54] code was employed for the AIMD simulations. Each trajectory was generated with a time step of 0.5 fs and 500 total steps. We train NequIP on 2500 reference structures sampled uniformly from the full data set of 6855 structures, use a validation set of 250 structures and evaluate the mean absolute error on all remaining structures. Due to the unbalanced nature of the data set (more atoms of Cu than in the molecule), we use a per-element weighed loss function in which atoms C, H, the sum of all O atoms, and the sum of all Cu atoms all receive equal weights. We weight the force term with $N_{atoms}^2 = 2,704$ and the energy term with 1.

$Li_4P_2O_7$ glass: The $Li_4P_2O_7$ ab-initio data were generated using an ab-initio melt-quench MD simulation, starting with a stoichiometric crystal of 208 atoms (space group P21/c) in a periodic box of $10.4 \times 14.0 \times 16.0$ Å. The dynamics used the Vienna Ab-Initio Simulation Package (VASP)[55–57], with a generalized gradient PBE functional[58], projector augmented wave (PAW) pseudopotentials[59], a NVT ensemble and a Nosé-Hoover thermostat, a time step of 2 fs, a plane-wave cutoff of 400 eV, and a Γ-point reciprocal-space mesh. The crystal melted at 3000 K for 50 ps, then immediately quenched to 600 K and run for another 50 ps. The resulting structure was confirmed to be amorphous by plotting the radial distribution function of P-P distances. The training was performed only on the molten portion, and the MD simulations for a quenched simulation. We sample the training sets uniformly from the full data set of 25,000 AIMD frames. We use a validation set of 100 structures, and evaluate the model on all remaining structures of the melt trajectory as well as on the full quench trajectory. The melt data were

shared with a previous study[13] and are available at https://doi.org/10.24433/CO.2788051.v1

LiPS: Lithium phosphorus sulfide (LiPS) based materials are known to exhibit high lithium ion conductivity, making them attractive as solid-state electrolytes for lithium-ion batteries. Other examples of known materials in this family of superionic conductors are LiGePS and LiCuPS-based compounds. The training data set is taken from a previous study on a graph neural network force field[13], where the LiPS training data were generated using ab-initio MD of an LiPS structure with Li-vacancy ($Li_{6.75}P_3S_{11}$) consisting of 27 Li, 12 P, and 44 S atoms respectively. The structure was first equilibrated and then run at 520 K using the NVT ensemble for 50 ps with a 2.0 fs time step. The full data set contains 25,001 MD frames. We choose training set sizes of 10, 100, 1000, and 2500 frames with a fixed validation set size of 100.

Liquid Water, Cheng et al.: The training set used in the data efficiency experiments on water consists of 1593 reference calculations of bulk liquid water at the revPBE0-D3 level of accuracy, with each structure containing 192 atoms, as given in[50]. Further information can be found in[50]. The data set was obtained from https://github.com/BingqingCheng/ab-initio-thermodynamics-of-water. We sample the training set uniformly from the full data set and for each experiment also use a validation set consisting of 100 structures. We then evaluate the error on a fixed hold-out test set of 190 structures.

Molecular Dynamics Simulations. To run MD simulations, NequIP force outputs were integrated with the Atomic Simulation Environment (ASE)[33] in which we implement a custom version of the Nosé-Hoover thermostat. We use this in-house implementation for the both the $Li_4P_2O_7$ as well as the LiPS MD simulations. The thermostat parameter was chosen to match the temperature fluctuations observed in the AIMD run. The RDF and ADFs for $Li_4P_2O_7$ were computed with a maximum distance of 6 Å (RDF) and 2.5 Å (both ADFs). The $Li_4P_2O_7$ MD simulations were started from the first frame of the AIMD quench simulation and the LiPS simulation was started from the first frame of the reference AIMD simulation of the corresponding training data.

Training. Networks are trained using a loss function based on a weighted sum of energy and a force loss terms:

$$\mathcal{L} = \lambda_E ||\hat{E} - E||^2 + \lambda_F \frac{1}{3N} \sum_{i=1}^{N} \sum_{\alpha=1}^{3} \left\| -\frac{\partial \hat{E}}{\partial r_{i,\alpha}} - F_{i,\alpha} \right\|^2 \quad (9)$$

where N is the number of atoms in the system, $\hat{E}$ is the predicted potential energy, and $\lambda_E$ and $\lambda_F$ are the energy- and force-weightings, respectively. While it is helpful to optimize the weightings as a hyperparameter, we found a relative weighting of energies to forces of 1 to $N_{atoms}^2$ a suitable default choice. Here the N accounts for the fact that that potential energy is a *global* quantity, while the atomic forces are *local* quantities and the square accounts for the fact that we use a MSE loss. This also makes the loss function size invariant. A full set of the weightings used in this work can be found in table 5.

We normalize the target energies by subtracting the mean potential energy over the training set and scale both the target energies and target force components by the root mean square of the force components over the training set. The predicted atomic energies $\hat{E}_i$ are scaled and shifted by two learnable per-species parameters before summing them for the total predicted potential energy $\hat{E}$:

$$\hat{E} = \sum_i \sigma_{s_i} \hat{E}_i + \lambda_{s_i} \quad (10)$$

where $\sigma_{s_i}$ and $\lambda_{s_i}$ are learnable per-species parameters indexed by $s_i$, the species of atom i. They are initialized to 1 and 0, respectively.

For the case of the joint training on water and ice, since the liquid water and ice structures have different numbers of atoms, we do not scale or shift the potential energy targets or force targets. Instead, we initialize the learnable per-species shift to the mean per-atom energy and initialize the learnable per-species scale to the average standard deviation over all force components in the training set.

Learning Curve Experiments. For learning curve experiments on the aspirin molecule in MD-17, a series of NequIP models with increasing order $l \in \{0, 1, 2, 3\}$ were trained on varying data set sizes. In particular, experiments were performed with a budget for training and validation of 200, 400, 600, 800, 1000 configurations, of which 50 samples were used for validation while the remaining ones were used for training. The reported test error was computed on the entire remaining MD-17 trajectory for each given budget. The weight-controlled version of NequIP was set up by creating a $l = 0$ network with increased feature size that matches the number of weights up to approx. 0.1% of the $l = 1$ network. The feature-controlled version of NequIP was set up by creating a $l = 0$ network with the same number of features as the $l = 1$ network, i.e. 4x more features than the original $l = 0$ network (1 scalar and 3 vector features), in particular the $l = 1$ network had a feature configuration of `64x0o + 64x0e + 64x1o + 64x1e` while the original $l = 0$ network used `64x0e` and feature-controlled $l = 0$ network used `512x0e`.

Hyperparameters. All models were trained on a NVIDIA Tesla V100 GPU in single-GPU training using float32 precision. For the small molecule systems, we use 5 interaction blocks, a learning rate of 0.01 and a batch size of 5. For the periodic systems, we use 6 interaction blocks, a learning rate of 0.005 and a batch size of 1. We decrease the initial learning rate by a decay factor of 0.8 whenever the validation loss in the forces has not seen an improvement for 50 epochs. We continuously save the model with the best validation loss in the forces and use the

model with the overall best validation loss for evaluation on the test set and MD simulations. For validation and test error evaluation, we use an exponential moving average of the training weights with weight 0.99. Training is stopped if either of the following conditions is met: (a) a maximum training time of of approximately seven days is reached; (b) a maximum number 1,000,000 epochs is reached; (c) the learning rate drops below $10^{-6}$; (d) the validation loss does not improve for 1000 epochs. We note that competitive results can typically be obtained within a matter of hours or often even minutes and most of the remaining training time is spent on only small improvements in the errors. We found the use of small batch sizes to be an important hyperparameter. We also found it important to choose the radial cutoff distance $r_c$ appropriately for a given system. In addition, we observed the number of layers to not have a strong effect as long as they were set within a reasonable range. We use different numbers of $l$ and feature dimensions for different systems and similarly also vary the cutoff radius for different systems. A full outline of the choices for $l$, feature size, cutoff radius as well as the weights for energies and forces in the loss function can be found in 5. All models were trained with both even and odd features. The weights were initialized according to a standard normal distribution (for details, see the e3nn software implementation[31]). The invariant radial networks act on a trainable Bessel basis of size 8 and were implemented with three hidden layers of 64 neurons with SiLU nonlinearities between them. The even scalars of the final interaction block are passed to the output block, which first reduces the feature dimension to 16 even scalars through a self-interaction layer. Finally, through another self-interaction layer, the feature dimension is reduced to a single scalar output value associated with each atom which is then summed over to give the total potential energy. Forces are obtained as the negative gradient of this predicted total potential energy, computed via automatic differentiation. All models were optimized with Adam with the AMSGrad variant in the PyTorch implementation[60–62] with $\beta_1 = 0.9$, $\beta_2 = 0.999$, and $\epsilon = 10^{-8}$ without weight decay. The average number of neighbors used for the $\frac{1}{\sqrt{N}}$ normalization of the convolution was computed over the full training set. For all molecular results, the average number of neighbors was computed once on the $N = 1000$ case for revised MD-17 and used for all other experiments. For the water sample efficiency and the LiPS experiments it was computed once on the N=1000 and N=2500 cases, respectively and then used for all other experiments for that system. The input files for training of NequIP models can be found at https://github.com/mir-group/nequip-input-files.

## Data availability

The Formate on Cu data set, the $Li_{6.75}P_3S_{11}$ data set, as well as the quench data for $Li_4P_2O_7$ have been deposited in the MaterialsCloud data base at https://doi.org/10.24435/materialscloud:s0-5n.

## Code availability

An open-source software implementation of NequIP is available at https://github.com/mir-group/nequip.

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

## Acknowledgements
We thank Jonathan Vandermause, Cheol Woo Park, David Clark, Kostiantyn Lapchevskyi, Joshua Rackers, and Benjamin Kurt Miller for helpful discussions. Work at Harvard University by S.B., L.S., N.M., and B.K. was supported by Bosch Research, the US Department of Energy, Office of Basic Energy Sciences Award No. DE-SC0022199 and the Integrated Mesoscale Architectures for Sustainable Catalysis (IMASC), an Energy Frontier Research Center, Award No. DE-SC0012573, by the NSF through the Harvard University Materials Research Science and Engineering Center Grant No. DMR-2011754, and by a Multidisciplinary University Research Initiative sponsored by the Office of Naval Research, under Grant N00014-20-1-2418. Work at Bosch Research by J.P.M. and M.K. was partially supported by ARPA-E Award No. DE-AR0000775 and used resources of the Oak Ridge Leadership Computing Facility at Oak Ridge National Laboratory, which is supported by the Office of Science of the Department of Energy under Contract DE-AC05-00OR22725. T.E.S. was supported by the Laboratory Directed Research and Development Program of Lawrence Berkeley National Laboratory and the Center for Advanced Mathematics for Energy Research Applications, both under U.S. Department of Energy Contract No. DE-AC02-05CH11231. M.G. was supported by a grant from the Simons Foundation (#454953 Matthieu Wyart). A.M is supported by U.S. Department of Energy, Office of Science, Office of Advanced Scientific Computing Research, Computational Science Graduate Fellowship under Award Number(s) DE-SC0021110. The authors acknowledge computing resources provided by the Harvard University FAS Division of Science Research Computing Group and by the Texas Advanced Computing Center (TACC) at The University of Texas at Austin under allocations DMR20009 and DMR20013.

## Author contributions
S.B. initiated the project, conceived the NequIP model, implemented the software and conducted all software experiments under the guidance of B.K. A.M. contributed to the development of the model and the software implementation. L.S. created the data set and helped with MD simulations of formate/Cu, and contributed to the development of the model and its software implementation. M.G. contributed to the development of the model and the software implementation. J.P.M. contributed to analyzing the LiPS conductor results and implemented the thermostat for MD simulations together with S.B. M.K. generated the AIMD data set of $Li_4P_2O_7$, wrote software for the analysis of MD results and contributed to the benchmarking on this system. N.M. wrote software for the estimation of diffusion coefficients and contributed to the interpretation of results. T.E.S. contributed to the conception of the model, guidance of computational experiments and software implementation. B.K. supervised the project from conception to design of experiments, implementation, theory, as well as analysis of data. All authors contributed to writing the manuscript.

## Competing interests
The authors declare no competing interests.
