## [Peer Review File · Nature Communications]

REVIEWER COMMENTS

Reviewer #1 (Remarks to the Author):

In this article the authors introduce NequIP, a SE(3)-Equivariant Graph Neural Networks for interatomic potentials' reconstruction. The method itself provides very good results even comparable with kernel methods on data efficiency. Nevertheless, the manuscript presents several issues that would require a major revision. Some of them are listed below.

1) The authors state: "However, such methods generally tend to exhibit poor computational scaling with the number of reference configurations, in both training (cubic in training set size) and prediction (linear in training set size). This limits both the amount of training data they can be trained on as well as the length and size of simulations that can be simulated with them."

This is a vague comment and should be backed up by numerical arguments, since in most cases what is important are the value of the prefactors. Meaning that a model can scale linear with N and still be slower than a N^4 scaling model for a given system. The authors should provide specific numerical values. Additionally, most ML force fields perform about the same, on the order of ms per sample.

2) It reads odd that what it is done in the article is introduced twice in the INTRODUCTION section (line 71 and line 170), and it is done again in line 198. I should advise to diversify the language.

3) In the comparison of MD17 performance shown in table I, indeed the NequIP shows indeed an amazing performance. Nevertheless, in the comparison to other approached including NequIP($l=0$) \approx SchNet, the discussion reads somehow flat (what is being told is just reading table I). Here, a more insightful discussion should be added, such as elaborating on the core differences between the compared methods and explain why a neural net of this nature can outperform even kernel methods.

4) Additionally, to support some claims in the paper, the authors should add the performance of the of SchNet in Fig. 7 as a reference. This will greatly help the narrative of the article.

5) Something that will immediately capture readers attention is why only forces are being compared against other models while in all the cited references there is always a comparison of both, total energies and forces. Furthermore, there is no discussion or arguments of why the authors only train the model on forces, while there are methods in the literature that even use energies and stress

tensors for training on top of forces. Everything needs to be justified. To address these issues, the comparison of energies and forces against other methods must be in the article, otherwise it could send the wrong message to the reader, meaning that the energy predictions are not accurate enough in this model (NequIP).

6) On the applications side, the authors show two interesting examples, Lithium Thiophosphate (LiPS) and Lithium Phosphate (Li₄P₂O₇). Table V summarizes the training performances for MAE of the forces. Interestingly, the learning capacity (according to this measure) is reached at 2500 training points for LiPS, converging to ~24 meV/Å. This is actually a good error compared to other reported methodologies. Nevertheless, it is well known that the MAE gives a good overview of the general performance of the model, but it softens the error implications of outliers. It would be more illustrative to present both performances, MAE and RMSE, given that comparing the differences between these two measures provide some degree of insights to monitor large errors on individual samples (local environments). Again, a complete evaluation of the learning performance of the method is not possible given that the energy errors are not reported.

7) Regarding the comparison between the dynamics using AIMD and NequIP-MD, Fig. 5 shows a really nice agreement. Just as a sanity check, I would add a simulation of NequIP-MD for a longer simulation time (500 ps, this should not be a problem for a ML model since they usually take a couple of ms per prediction as shown in Table VI) just to be sure that the simulation remains stable and reliable. Then report its $g(r)$ in Fig. 5 as comparison. I understand that they could differ a bit.

8) In the case of Li₄P₂O₇, I cannot give a proper evaluation of the application if the RMSE are missing but more importantly, because of the missing energy errors. The energetics in diffusion is crucial, and because of this, I'm not sure how to judge the reported diffusivity values. Furthermore, this application needs a convergence analysis (diffusivity vs length of the simulations) since 50 ps is an extremely short time for such a macroscopic variable. Hence, in order to discard any possible artefacts of the NequIP, such analysis should be provided.

9) The results provided in Fig. 7 are very revealing and show the great impact of using equivariant internal features. It would give a further insight on the efficacy of the model if also $l=2$ is included in this example. This should definitively evince the need of such architectures in force fields reconstruction.

10) The DISCUSSION section, does not present any discussion, it is more like a summary or conclusions. This section should be intended to present an insightful but compact discussion of the current state-of-the-art of the field and why and how your methodology actually advances the field (ML force fields) and what practitioners would gain if they use it.

11) In more stylistic issues, the reference style is not appropriate. Some of them are not updated.

In general, the reported method provides very good results clearly stating what is needed to be included in NN-based interatomic potentials. Nevertheless, in its current state I advise to not accept the manuscript. This based in several reasons, firstly, the method itself or the ideas behind are not new. Similar ideas were developed by Gastegger et al. (arXiv:2010.14942, 2020) and more recently by Schütt et al. (arXiv:2102.03150, 2021)**, but more importantly, this work is a minor/incremental development on top of tensor field networks (2018). This is not necessarily a bad thing but then has to be compensated by presenting insightful discussions and analysis, which currently is missing.

At the moment, the presentation of the article is more appropriate for a computational physics/chemistry journal, given that it introduces the method (which mostly was already introduced in previous publications), then present comparisons to known methods and datasets (no insightful or revealing arguments are given only numerics), and then applications, which are very interesting but need further analysis to show the reliability of the method.

The biggest concern is the fact that the energy part of the model is missing, even though the main predictor of the network is indeed the energy (eq. 3). Furthermore, total energy analysis is a standard measure of accuracy in the field, hence this shows that the work is not complete.

**This article appeared on arXiv ~two weeks after NequIP and introduces a method that is practically the same (only some technical parts are different but the final coded expressions follow the same idea and both truncate the spherical harmonics at $L=1$). This should not be considered as a bad point since they were developed independently, and they show the same performances.

Reviewer #2 (Remarks to the Author):

This paper introduces a new framework (NequIP) for constructing machine learning potentials (MLPs), using a rotational-symmetry-equivariant neural network combined with graph convolution. The paper then shows several benchmarks on diverse systems including small molecular datasets, bulk water, organic molecules on a surface, and a superionic conductor. The approach is interesting as well as provocative, and the example applications are convincing.

A few questions, most are concerning presentations:

* GPR/KRR are often combined with sparsification in order to scale better with the number of data points. This point is ignored in the paper.

* Eqns.2-4 are rather obvious and can be condensed.

* I am a bit confused by Eqn.5: does it mean that the interacting force vector between two atoms i and j is only dependent on the displacement r_{ij} ? But one would expect that the force also depends on the other atoms close to i and j . Presumably such many-body effects are considered during the convolution step, but this is not clear from Eqn.5.

* Page 9, line 446: The authors wrote "ability to achieve high accuracy on a comparatively small data set opens the door to training models on expensive high-order ab-initio quantum chemical methods." This seems to be a bit overstated, as previous MLPs that trained on CCSD(T) and QMC already exist.

* Page 10, line 493: What does it mean to "sampled uniformly"? Does it mean taking a snapshot from every X step of the MD trajectory, or does it mean a random selection, or a uniform sampling in the space of the design matrix?

* Although it is very impressive that the authors are able to train a MLP for water with 1000x fewer training data compared to DeepMD, bear in mind that the first generation of DeepMD (the unsmooth version) is highly data-inefficient due to the "cusps" introduced by the permutations of atoms. Also the water data set contains the whole MD trajectory, meaning that the training data is highly correlated so significant sparsification can be done before hampering the amount of information contained in the data set. I suggest that the authors consider including these more nuanced points, so that readers who are not familiar with the field won't be misled.

* Some discussions on how the atomic cutoffs and the number of convolutions affect the results can be very helpful to readers.

Minor point:

* Perhaps better to replot the gor in the left panel of Fig.5 using a xrange of [1,6] Angstroms.

Bingqing Cheng

Reviewer #3 (Remarks to the Author):

This paper reports an important advance in the development in the area of machine learning interatomic potentials. In particular, by building an equivariant framework that can act on tensors of different order (scalar, vector, high-order tensors), significant improvements in accuracy for smaller training data sets can be achieved, thus addressing a significant challenge in the construction of ML potentials. As such, I believe this paper will be well-received by the community. I have no major criticisms of this paper and the way it is presented, but I would like to make two suggestions:

1. Presenting tables of MAEs is useful to see the performance of the proposed ML scheme, but I think the reader needs to see some representation of the training data. I realize this might not be easy given the many features captured in the data, but an example could be showing a scatter plot of force magnitudes (since training is done using forces, as suggested in Eq. (9)) or even corresponding energies in the MD-17 data set as a function of some key geometric features [the authors might want to see the recent work of Bogojeski et al. Nature Comm. 11, 5223 (2020) for an example of what could be done]. A few selected molecules could be chosen for the main paper and the rest put into a supplementary information document. This might already be sufficient, although if they can find a convenient and visually intuitive representation of the training data for the extended systems, that would be even better.

2. The idea of learnable radial functions in the decomposition of the convolution filters presented in Eq. (5) is an interesting one. Could the authors provide some justification for why it is necessary to learn these using a small NN as opposed to simply expanding them in a basis set or as a polynomial and then learning the coefficients of each as part of the overall training scheme. This idea has been investigated for learning activation functions in neural networks [see, for example, Goyal, et al. 101, 1-18 (2019)]. Wouldn't this be an easier scheme overall?

E(3)-Equivariant Graph Neural Networks for Data-Efficient and Accurate Interatomic Potentials

Simon Batzner[†], Albert Musaelian[†], Lixin Sun[†], Mario Geiger[¶], Jonathan P. Mailoa[‡], Mordechai Kornbluth[‡], Nicola Molinari[†], Tess Smidt^{*§}, and Boris Kozinsky^{†‡}

[†] John A. Paulson School of Engineering and Applied Sciences, Harvard University, Cambridge, MA 02138, USA

[¶] École Polytechnique Fédérale de Lausanne, 1015 Lausanne, Switzerland

^{*} Computational Research Division and Center for Advanced Mathematics for Energy Research Applications, Lawrence Berkeley National Laboratory, Berkeley, CA 94720, USA

[§]Massachusetts Institute of Technology, Department of Electrical Engineering and Computer Science, Cambridge, MA 02142, USA

[‡] Robert Bosch Research and Technology Center, Cambridge, MA 02139, USA

Response to the reviewers, Nature Communications NCOMMS-21-05986A

We appreciate the time and effort that the editor and the three reviewers have invested in reviewing our manuscript. We address the comments and questions raised by each reviewer below. The green text refers to what was present in the old version of the manuscript, while the blue text indicates changes or additions to the revised manuscript/supplemental material.

Reviewer 1

Reviewer: In this article the authors introduce NequIP, a SE(3)-Equivariant Graph Neural Networks for interatomic potentials’ reconstruction. The method itself provides very good results even comparable with kernel methods on data efficiency. Nevertheless, the manuscript presents several issues that would require a major revision. Some of them are listed below.

Author reply: We thank the Referee for the review. In the text below, we answer the questions raised point-by-point.

Reviewer: 1) The authors state: “However, such methods generally tend to exhibit poor computational scaling with the number of reference configurations, in both training (cubic in training set size) and prediction (linear in training set size). This limits both the amount of training data they can be trained on as well as the length and size of simulations that can be simulated with them.” This is a vague comment and should be backed up by numerical arguments, since in most

cases what is important are the value of the prefactors. Meaning that a model can scale linear with N and still be slower than a N^4 scaling model for a given system. The authors should provide specific numerical values. Additionally, most ML force fields perform about the same, on the order of ms per sample.

Author reply: We thank the referee for this comment. We agree, that the prefactor certainly matters in determining the overall computational efficiency of the method which may vary across systems. The discussion on the scaling of kernel methods is not critical to the issue of sample efficiency of the presented approach and we have therefore removed the following corresponding lines from the manuscript:

Original:

Kernel-based approaches, such as e.g. Gaussian Processes (GP) [1, 2] or Kernel Ridge Regression (KRR) [3], are a way to remedy this problem as they often generalize better from limited sample sizes. However, such methods generally tend to exhibit poor computational scaling with the number of reference configurations, in both training (cubic in training set size) and prediction (linear in training set size). This limits both the amount of training data they can be trained on as well as the length and size of simulations that can be simulated with them.

Reviewer: 2) It reads odd that what it is done in the article is introduced twice in the INTRODUCTION section (line 71 and line 170), and it is done again in line 198. I should advise to diversify the language.

Author reply: We have diversified the language accordingly in the following two places. We note that we have also added equivariance with respect to parity, making the approach equivariant to the Euclidean group $E(3)$, instead of $SE(3)$ and have adapted the text accordingly:

Original:

In this work we present a deep learning energy-conserving interatomic potential for both molecules and materials built on $SE(3)$ -equivariant convolutions over geometric tensors that yields state-of-the-art accuracy, outstanding data-efficiency, and can with high fidelity reproduce structural and kinetic properties from molecular dynamics simulations.

Changed to:

The contribution of the present work is the introduction of a deep learning energy-conserving interatomic potential for both molecules and materials built on $E(3)$ -equivariant convolutions over geometric tensors that yields state-of-the-art accuracy, outstanding data-efficiency, and can with high fidelity reproduce structural and kinetic properties from molecular dynamics simulations.

Original:

In this work, we focus on equivariance with respect to $SE(3)$, i.e. the group of rotations and translations in 3D space.

Changed to:

Here, we focus on equivariance with respect to $E(3)$, i.e. the group of rotations, reflections, and translations in 3D space.

Reviewer: 3) In the comparison of MD17 performance shown in table I, indeed the NequIP shows indeed an amazing performance. Nevertheless, in the comparison to other approached including NequIP($l=0$)SchNet, the discussion reads somehow flat (what is being told is just reading table I). Here, a more insightful discussion should be added, such as elaborating on the core differences between the compared methods and explain why a neural net of this nature can outperform even kernel methods.

Author reply: We thank the referee for this comment. We have revised the discussion of the results on MD-17 to elaborate more on how NequIP is different from other methods. We have also updated the tables to include data on different rotations orders l for NequIP and have added a table on results on the revised MD-17 data set since the revised version contains less noisy reference data:

Original:

We first evaluate NequIP on MD-17 [3, 4, 5], a data set of eight small organic molecules in which reference values of energy and forces are generated by ab-initio MD simulations with DFT. For training we use $N=1,000$ structure configurations for each molecule, sampled uniformly from the full data set, the same number of configurations for validation, and evaluate the test error on all remaining configurations in the data set. The mean absolute error in the force components is shown in Table I in units of $[\text{meV}/\text{\AA}]$. We compare results using NequIP with those from published leading ML-IP models that were also trained on 1,000 structures: in particular SchNet [6], DimeNet [7] (both graph neural networks), sGDML [5], and FCHL19/GPR (kernel-based methods) [8]. We find that NequIP outperforms SchNet and sGDML on all molecules in the data set, DimeNet on 7 out of 8 molecules (on par on the remaining one), and performs on par with FCHL/19GPR. The consistent improvement in accuracy upon sGDML and the comparable performance to FCHL19/GPR are particularly surprising, as these are based on kernel methods, that typically tend to be more sample efficient. It should be noted, however, that the evaluation cost of kernel methods scales linearly with the number of training configurations. Note also that on some molecules, NequIP trained on 1,000 configurations even performs as well as SchNet trained on 50,000 structures [6]: on aspirin and naphthalene, for example, the NequIP network trained on 1,000 structures produces mean absolute errors in the forces of $15.1 \text{ meV}/\text{\AA}$ and $4.2 \text{ meV}/\text{\AA}$, respectively, compared to $14.3 \text{ meV}/\text{\AA}$ and $4.8 \text{ meV}/\text{\AA}$ of SchNet trained on 50x more molecules, hinting that NequIP exhibits exceptional data efficiency. On other molecules such as ethanol, however, SchNet trained with 50,000 molecules still clearly outperforms NequIP trained with 1,000 molecules

(2.2 meV/Å for SchNet for N=50,000 vs 9.0 meV/Å for NequIP for N=1,000).

Changed to:

We first evaluate NequIP on MD-17 [3, 4, 5], a data set of small organic molecules in which reference values of energy and forces are generated by ab-initio MD simulations with DFT. Recently, a recomputed version of the original MD-17 data with higher numerical accuracy has been released, termed the revised MD-17 data set [9] (an example histogram of potential energies and force components can be found in Appendix C). In order to be able to compare results to a wide variety of methods, we benchmark NequIP on both data sets. For training and validation, we use a combined N=1,000 configurations. The mean absolute error in the energies and force components is shown in Tables I and II. We compare results using NequIP with those from published leading MLIP models. We find that NequIP significantly outperforms invariant GNN-IPs (such as SchNet [6] and DimeNet [7]), shallow neural networks (such as ANI [10]), and kernel-based approaches (such as GAP [1], FCHL19/GPR [8, 9] and sGDML [5]). Finally, we compare to a series of other methods including ACE [11], SpookyNet [12], and GemNet [13] as well as other equivariant neural networks such as PaiNN [14], NewtonNet [15] and UNiTE [16]. Again, it should be stressed that PaiNN and NewtonNet are $l_{max} = 1$ -only versions of equivariant networks. The results for ACE, GAP, and ANI on the revised MD-17 data set are those reported in [17]. Importantly, we train and test separate NequIP models on both the original and the revised MD-17 data set, and find that NequIP obtains significantly lower energy errors on the revised data set, while the force accuracy is similar on the two data sets. In line with previous work [9], this suggests that the noise floor on the original MD-17 data is higher on the energies and that only the results on the revised MD-17 data set should be used for comparing different methods. Remarkably, we find that NequIP outperforms all other methods. The consistent improvement in accuracy compared to sGDML and FCHL19/GPR are particularly surprising, as these are based on kernel methods, which are typically more sample efficient, but whose evaluation cost scales linearly with the number of training configurations. We run a convergence scan on the rotation order $l \in \{0, 1, 2, 3\}$ and find that increasing the tensor rank beyond $l = 1$ gives a consistent improvement. The significant improvement from $l = 0$ to $l = 1$ highlights the crucial role of equivariance in obtaining improved accuracy on this task.

Reviewer: 4) Additionally, to support some claims in the paper, the authors should add the performance of the of SchNet in Fig. 7 as a reference. This will greatly help the narrative of the article.

Author reply: We note that a $l = 0$ NequIP network is approximately equivalent to a SchNet network, as both are message-passing neural networks that use the same types of scalar features and scalar operations. Training a series of SchNet networks would require optimizing the SchNet hyperparameters for this system and as

external users we would be likely to achieve suboptimal results. We show the $l = 0$ network in this graph and note this equivalence in the text:

Original:

Second, when all interactions involving $l = 1$ are turned off, this turns the network into a conventional invariant GNN-IP, involving only invariant convolutions over scalar features in a SchNet-style fashion.

Reviewer: 5) Something that will immediately capture readers attention is why only forces are being compared against other models while in all the cited references there is always a comparison of both, total energies and forces. Furthermore, there is no discussion or arguments of why the authors only train the model on forces, while there are methods in the literature that even use energies and stress tensors for training on top of forces. Everything needs to be justified. To address these issues, the comparison of energies and forces against other methods must be in the article, otherwise it could send the wrong message to the reader, meaning that the energy predictions are not accurate enough in this model (NequIP).

Author reply: We thank the reviewer for this important comment. We have updated our model to also be trained on energies and are reporting energies for every experiment conducted in this work. All tables of results have been changed accordingly. Furthermore, we have updated the loss function and the discussion accordingly.

Original:

Networks are trained using a loss function based on atomic forces:

$$\mathcal{L} = \frac{1}{3N} \sum_{i=1}^N \sum_{\alpha=1}^3 \left\| -\frac{\partial \hat{E}}{\partial r_{i,\alpha}} - F_{i,\alpha} \right\|^2$$

where N is the number of atoms in the system and \hat{E} is the predicted potential energy. Note that we do not train on energies since atomic forces are the only quantities required to integrate Newton’s equations of motion. Since the predicted forces are computed as the gradient of a scalar potential, they are still conservative. If energies are of interest, however, one can add them to the loss function and determine the relative weighting via a trade-off parameter as done in previous works [6, 7]. In a similar fashion, it is trivial to add other quantities of interest to the loss function (e.g predicting atomic charges or multipole tensors can be of interest for modeling long-range interactions), where they may be scalar fields, vector fields, or higher-order tensor fields.

Changed to:

Networks are trained using a loss function based on a weighted sum of energy

and a force loss terms:

$$\mathcal{L} = \lambda_E \|\hat{E} - E\|^2 + \lambda_F \frac{1}{3N} \sum_{i=1}^N \sum_{\alpha=1}^3 \left\| -\frac{\partial \hat{E}}{\partial r_{i,\alpha}} - F_{i,\alpha} \right\|^2$$

where N is the number of atoms in the system, \hat{E} is the predicted potential energy, and λ_E and λ_F are the energy- and force-weightings, respectively. While it is helpful to optimize the weightings as a hyperparameter, we found a relative weighting of energies to forces of 1 to N_{atoms}^2 a suitable default choice. Here the N accounts for the fact that that potential energy is a *global* quantity, while the atomic forces are *local* quantities and the square accounts for the fact that we use a MSE loss. This also makes the loss function size invariant. A full set of the weightings used in this work can be found in table VII. We normalize the target energies by subtracting the mean potential energy over the training set and scale both the target energies and the target force components by the root mean square of the force components over the training set. The predicted atomic energies \hat{E}_i are scaled and shifted by two learnable per-species parameters before summing them for the total predicted potential energy \hat{E} :

$$\hat{E} = \sum_i \sigma_{s_i} \hat{E}_i + \lambda_{s_i}$$

where σ_{s_i} and λ_{s_i} are learnable per-species parameters indexed by s_i , the species of atom i . They are initialized to 1 and 0, respectively. For the case of the joint training on water and ice, since the liquid water and ice structures have different numbers of atoms, we do not scale or shift the potential energy targets or force targets. Instead, we initialize the learnable per-species shift to the mean per-atom energy and initialize the learnable per-species scale to the average standard deviation over all force components in the training set.

Reviewer: 6) On the applications side, the authors show two interesting examples, Lithium Thiophosphate (LiPS) and Lithium Phosphate (Li4P2O7). Table V summarizes the training performances for MAE of the forces. Interestingly, the learning capacity (according to this measure) is reached at 2500 training points for LiPS, converging to 24 meV/Å. This is actually a good error compared to other reported methodologies. Nevertheless, it is well known that the MAE gives a good overview of the general performance of the model, but it softens the error implications of outliers. It would be more illustrative to present both performances, MAE and RMSE, given that comparing the differences between these two measures provide some degree of insights to monitor large errors on individual samples (local environments). Again, a complete evaluation of the learning performance of the method is not possible given that the energy errors are not reported.

Author reply: We thank the reviewer for this important comment. We have update the tables accordingly and have added both the RMSE of the forces. as well as MAE and

RMSE on the energies. Due to the addition of higher-order tensors, we also find a significant reduction in test errors.

Reviewer: 7) Regarding the comparison between the dynamics using AIMD and NequIP-MD, Fig. 5 shows a really nice agreement. Just as a sanity check, I would add a simulation of NequIP-MD for a longer simulation time (500 ps, this should not be a problem for a ML model since they usually take a couple of ms per prediction as shown in Table VI) just to be sure that the simulation remains stable and reliable. Then report its $g(r)$ in Fig. 5 as comparison. I understand that they could differ a bit.

Author reply: We thank the reviewer for this comment. We have added a 500 ps simulation of the LiPO system and compare to the the 50ps AIMD simulation. We find that the $g(r)$ predicted by the 500ps NequIP-driven simulation matches the AIMD simulation well. We have added this in the appendix as follows:

Changed to:

Figure 9 shows the Radial Distribution Function obtained from a MD simulation of simulation length 500ps compared to the AIMD simulation of 50ps. For both simulations, there first 10ps were not used in the computation of the RDF.

We have also added a reference to it in the main text:

Changed to:

We also include results from a longer NequIP-driven MD simulation of 500 ps, which can be found in Appendix A.

Reviewer: 8) In the case of Li₄P₂O₇, I cannot give a proper evaluation of the application if the RMSE are missing but more importantly, because of the missing energy errors. The energetics in diffusion is crucial, and because of this, I'm not sure how to judge the reported diffusivity values. Furthermore, this application needs a convergence analysis (diffusivity vs length of the simulations) since 50 ps is an extremely short time for such a macroscopic variable. Hence, in order to discard any possible artefacts of the NequIP, such analysis should be provided.

Author reply: We believe the reviewer here meant to refer to the Lithium Thiophosphate (LiPS) system here, since no experiments on diffusion in Li₄P₂O₇ were performed. Again, we have added the RMSE of the forces, as well as both MAE and RMSE of the energies. We note that Li_{6.75}P₃S₁₁ is a good conductor and therefore a simulation length of 50ps at a high temperature of $T = 520K$ is not short, a simulation length of 30ps even would be sufficient to obtain good values for the ionic conductivity. Two examples in the literature are [18] and [19], which perform AIMD simulations of a lengths of 40ps and 60 ps, respectively, in systems with similar compositions. Finally, we note that for the runs we used to compute the diffusivity, both AIMD and NequIP display a log-log

slope of approximately 1, meaning that the simulations are in the Fick’s Law diffusive regime. We have updated the text as follows:

Original:

Here we demonstrate that not only does NequIP obtain small errors in the force components, but it also accurately predicts the diffusivity after training on a data set obtained from an AIMD simulation. Again, we find that very small training sets lead to highly accurate models, as shown in Table V for training set sizes of 10, 100, 1,000 and 2,500 structures. We run a series of MD simulations with the NequIP potential trained on 2,500 structures in the NVT ensemble at the same temperature as the AIMD simulation for a total simulation time of 50 ps and a time step of 0.25 fs, which we found advantageous for reliability and stability of long simulations. We measure the Li diffusivity in ten NequIP-driven MD simulations (computed via the slope of the mean square displacement), all of length 50 ps and started from different initial velocities, randomly sampled from a Maxwell-Boltzmann distribution. We find a mean diffusivity of $1.42 \times 10^{-5} \text{ cm}^2/\text{s}$, in excellent agreement with the diffusivity of $1.38 \times 10^{-5} \text{ cm}^2/\text{s}$ computed from AIMD, thus achieving a relative error of as little as 3%. Figure 6 shows the mean square displacements of Li for an example run.

Changed to:

Here we demonstrate that not only does NequIP obtain small errors in the energies and force components, but it also accurately predicts the diffusivity after training on a data set obtained from an AIMD simulation. Again, we find that very small training sets lead to highly accurate models, as shown in Table VI for training set sizes of 10, 100, 1,000 and 2,500 structures. We run a series of MD simulations with the NequIP potential trained on 2,500 structures in the NVT ensemble at the same temperature as the AIMD simulation for a total simulation time of 50 ps and a time step of 0.25 fs, which we found advantageous for the reliability and stability of long simulations. We measure the Li diffusivity in these NequIP-driven MD simulations (computed via the slope of the mean square displacement) started from different initial velocities, randomly sampled from a Maxwell-Boltzmann distribution. We find a mean diffusivity of $1.25 \times 10^{-5} \text{ cm}^2/\text{s}$, in excellent agreement with the diffusivity of $1.37 \times 10^{-5} \text{ cm}^2/\text{s}$ computed from AIMD, thus achieving a relative error of as little as 9%. Figure 6 shows the mean square displacements of Li for an example run of NequIP in comparison to AIMD.

Reviewer: 9) The results provided in Fig. 7 are very revealing and show the great impact of using equivariant internal features. It would give a further insight on the efficacy of the model if also $l=2$ is included in this example. This should definitively evince the need of such architectures in force fields reconstruction.

Author reply: We thank the reviewer for this positive comment. We have conducted an analysis of the behavior of this learning curves as a function of l and have included

results on $l \in \{0, 1, 2, 3\}$. In particular, we have added to the discussion of the change in slope in the log-log plot of error vs training set size. The text has been updated to discuss this behavior. We have also added experiments that control for the different number of weights and features that come with using a higher-order tensor and have added a comparison to existing method to demonstrate the change in log-log slope observed for NequIP training.

Original:

In the above experiments, NequIP exhibits exceptionally high data efficiency, i.e. it can be trained successfully to state-of-the-art accuracy from unexpectedly small training sets. It is interesting to consider the reasons for such high performance and verify that it is connected to the equivariant nature of the model. First, it is important to note that each training configuration contains multiple labels, thus increasing the total number of labels available beyond just the potential energy label associated with each structure. In particular, for a training set of M first-principles calculations with structures consisting of N atoms, the total number of labels available is $M(3N + 1)$ since every force component on every atom constitutes a label and so does the total energy of the reference calculation (we only train to atomic forces and not energies, thus using $3MN$ force components as labels). In order to gain insight into the reasons behind increased accuracy and data efficiency, we perform a series of experiments with the goal of isolating the effect of using equivariant convolutions of geometric tensors compared to invariant convolutions over scalars. In particular, we run a set of experiments for a system with a fixed number of training configurations in which we explicitly turn on or off interactions of higher order than $l = 0$. This defines two settings: first, we train the network with both $l = 0$ and $l = 1$ features and all five interactions as previously outlined in this work. Second, when all interactions involving $l = 1$ are turned off, this turns the network into a conventional invariant GNN-IP, involving only invariant convolutions over scalar features in a SchNet-style fashion. As a test system we chose bulk water: in particular we use the data set introduced in [20], consisting of 1,593 bulk liquid water structures with 64 water molecules each. We train a series of networks with identical hyperparameters, but vary the training set sizes between 10 and 1,000 structures, sampled uniformly from the full data set, as well as a validation set consisting of 100 structures. We then evaluate the error on all remaining structures for a given training set size. As shown in Figure 7, we find that the equivariant setting (using $l = 0$ and $l = 1$) significantly outperforms the invariant setting (using only $l = 0$) for all data set sizes as measured by the MAE of force components. This suggests that it is indeed the use of tensor (in our specific case vector) features and equivariant convolutions that enables the high data efficiency of NequIP. We further note, that in [20], a Behler-Parrinello Neural Network (BPNN) was trained on 1303 structures, yielding a RMSE of ≈ 120 meV/Å in forces when evaluated on the remaining 290 structures. We find that NequIP models trained with as little

as 50 and 100 data points obtain RMSEs of 122.9 meV/Å and 93.3 meV/Å on their respective test sets (note that Figure 7 shows the MAE). This provides further evidence that NequIP exhibits significantly improved data efficiency in comparison with existing methods.

Changed to:

In the above experiments, NequIP exhibits exceptionally high data efficiency. It is interesting to consider the reasons for such high performance and verify that it is connected to the equivariant nature of the model. First, it is important to note that each training configuration contains multiple labels: in particular, for a training set of M first-principles calculations with structures consisting of N atoms, the energies and force components together give a total of $M(3N + 1)$ labels. In order to gain insight into the reasons behind increased accuracy and data efficiency, we perform a series of experiments with the goal of isolating the effect of using equivariant convolutions. In particular, we run a set of experiments in which we explicitly turn on or off interactions of higher order than $l = 0$. This defines two settings: first, we train the network with the full set of tensor features up to a given order l and the corresponding equivariant interactions. Second, we turn off all interactions involving $l > 0$, making the network a conventional invariant GNN-IP, involving only invariant convolutions over scalar features in a SchNet-style fashion. As a first test system we choose bulk water: in particular we use the data set introduced in [20]. We train a series of networks with identical hyperparameters, but vary the training set sizes between 10 and 1,000 structures. As shown in Figure 7, we find that the equivariant networks with $l \in 1, 2, 3$ significantly outperform the invariant networks with $l = 0$ for all data set sizes as measured by the MAE of force components. This suggests that it is indeed the use of tensor features and equivariant convolutions that enables the high sample efficiency of NequIP. In addition, it is apparent that the learning curves of equivariant networks have a different slope in log-log space. It has been observed that learning curves typically follow a power-law of the form [21]: $\epsilon \propto aN^b$ where ϵ and N refer to the generalization error and the number of training points, respectively. The exponent of this power-law (or equivalently the slope in log-log space) determines how fast a learning algorithm learns as new data become available. Empirical results have shown that this exponent typically remains fixed across different learning algorithms for a given data set, and different methods only *shift* the learning curve, leaving the log-log slope unaffected [21]. The same trend can also be observed for various methods on the aspirin molecule in the MD-17 data set (see Figure 8) where across a series of descriptors and regression models (sGDML, FCHL19, and PhysNet [5, 8, 22]) the learning curves show an approximately similar log-log slope (results obtained from <http://quantum-machine.org/gdml/#datasets>). To our surprise, we observe that the equivariant NequIP networks break this pattern. Instead they follow a log-log slope with larger magnitude, meaning that they learn faster

as new data become available. An invariant $l = 0$ NequIP network, however, displays a similar log-log slope to other methods, suggesting that it is indeed the equivariant nature of NequIP that allows for the change in learning behavior. Further increasing the rotation order l beyond $l = 1$ again only shifts the learning curve and does not result in an additional change in log-log slope. To control for the different number of weights and features in orders of different rotation order l , we report weight- and feature-controlled data in Appendix B. Both show qualitatively the same effect. The Appendix also contains results on the behavior of the energies, when trained jointly with forces. For details on the training setup and the control experiments, see the Methods section. We further note, that in [20], a Behler-Parrinello Neural Network (BPNN) was trained on 1303 structures, yielding a RMSE of ≈ 120 meV/Å in forces when evaluated on the remaining 290 structures. We find that NequIP $l = 2$ models trained with as little as 100 and 250 data points obtain RMSEs of 129.8 meV/Å and 103.4 meV/Å respectively (note that Figure 7 shows the MAE). This provides further evidence that NequIP exhibits significantly improved data efficiency in comparison with existing methods.

Reviewer: 10) The DISCUSSION section, does not present any discussion, it is more like a summary or conclusions. This section should be intended to present an insightful but compact discussion of the current state-of-the-art of the field and why and how your methodology actually advances the field (ML force fields) and what practitioners would gain if they use it.

Author reply: We thank the referee for this note. We have updated the DISCUSSION section accordingly:

Original:

We demonstrate that the Neural Equivariant Interatomic Potential (NequIP), a new type of graph neural network built on SE(3)-equivariant convolutions exhibits state-of-the-art accuracy and exceptional data efficiency on data sets of small molecules and periodic materials. Furthermore, we find that we can reproduce structural and kinetic properties from molecular dynamics simulations with very high fidelity in comparison to ab-initio simulations. The ability to both learn from small numbers of reference samples, while retaining high computational efficiency opens the door to performing simulations of large systems over long time-scales at quantum mechanical accuracy, using DFT or higher order methods such as coupled-cluster or quantum Monte Carlo data as reference. We expect the new method will enable researchers in computational chemistry, physics, biology, and materials science to conduct molecular dynamics simulations of complex reactions and phase transformations at increased accuracy and efficiency.

Changed to:

We demonstrate that the Neural Equivariant Interatomic Potential (NequIP),

a new type of graph neural network built on E(3)-equivariant convolutions, exhibits state-of-the-art accuracy and exceptional data efficiency on data sets of small molecules and periodic materials. We isolate that the improvements are due to the introduction of equivariant representations in place of more widely used invariant representations. This raises questions about the optimal way to include symmetry in Machine Learning for molecules and materials. A better understanding of why equivariance enables improved sample efficiency and accuracy is likely to be a fruitful direction towards designing better ML algorithms for the construction of Potential Energy Surfaces. In addition to open questions around the effect of equivariance on accuracy and learning dynamics, a clear theoretical understanding of how the many-body character of interactions arises in Message Passing Interatomic Potentials remains elusive. We expect the proposed method will enable researchers in computational chemistry, physics, biology, and materials science to conduct molecular dynamics simulations of complex reactions and phase transformations at increased accuracy and efficiency.

Reviewer: In general, the reported method provides very good results clearly stating what is needed to be included in NN-based interatomic potentials. Nevertheless, in its current state I advise to not accept the manuscript. This based in several reasons, firstly, the method itself or the ideas behind are not new. Similar ideas were developed by Gastegger et al. (arXiv:2010.14942, 2020) and more recently by Schütt et al. (arXiv:2102.03150, 2021)**, but more importantly, this work is a minor/incremental development on top of tensor field networks (2018). This is not necessarily a bad thing but then has to be compensated by presenting insightful discussions and analysis, which currently is missing.

**This article appeared on arXiv two weeks after NequIP and introduces a method that is practically the same (only some technical parts are different but the final coded expressions follow the same idea and both truncate the spherical harmonics at $L=1$). This should not be considered as a bad point since they were developed independently, and they show the same performances.

Author reply: We acknowledge that indeed a similar idea to NequIP appeared online 4 weeks after the present work first was posted (NequIP was posted on Jan 5, the work from Schütt et al. 28 days later on Feb 8). As stated by the referee, the works were developed independently and the ideas proposed in Schütt et al. can be seen as an $l = 1$ -only subset of NequIP. As outlined in the present work, however, we find it important to be able to include higher-order interactions in the network.

The work by Gastegger et al. introduces FieldSchNet, an approach for modeling interactions of molecules with external fields. While this work includes using vector-based features, the interactions are inspired by modeling dipole-field and dipole-dipole interactions and are therefore special instances that do

not constitute a fully equivariant neural network as outlined in the present work. Furthermore, to the best of our knowledge, FieldSchNet does not implement the $1 \otimes 1 \rightarrow 1$ interaction, which a conventional $l = 1$ equivariant network would introduce. These interactions alone therefore only have limited similarity to the method proposed in the present work.

Regarding the need for insightful discussions and analysis asked for by the reviewer, we have shown a detailed analysis of the role of the rotation order l , its impact on accuracy, and in particular the unique effect of equivariance on learning curves, which to the best of our knowledge had not been known prior to this work.

We further note that while the original Tensor Field Networks paper discussed the more general idea of equivariant neural networks and laid the theoretical foundation, the work presented here implements simulations on realistic chemical systems, demonstrates state-of-the-art accuracy and sample-efficiency, quantifies the effect of equivariance on learning curves, applications to periodic systems, and ablation studies on the effect of equivariance vs. invariance on these, all of which were not studied in the original TFN work. In addition, substantial work was required to generalize and optimize the TFN code for general $E(3)$ -equivariant neural networks. For example, the original code only went up to $l = 1$, whereas here we can go up to arbitrary l .

Reviewer: The biggest concern is the fact that the energy part of the model is missing, even though the main predictor of the network is indeed the energy (eq. 3). Furthermore, total energy analysis is a standard measure of accuracy in the field, hence this shows that the work is not complete.

Author reply: We thank the reviewer for this important comment. As discussed, we have updated every experiment shown in the manuscript with total energy errors.

Reviewer 2

Reviewer: This paper introduces a new framework (NequIP) for constructing machine learning potentials (MLPs), using a rotational-symmetry-equivariant neural network combined with graph convolution. The paper then shows several benchmarks on diverse systems including small molecular datasets, bulk water, organic molecules on a surface, and a superionic conductor. The approach is interesting as well as provocative, and the example applications are convincing. A few questions, most are concerning presentations:

Author reply: We thank the reviewer for this positive review. All comments are addressed below one-by-one.

Reviewer: * GPR/KRR are often combined with sparsification in order to scale better with the number of data points. This point is ignored in the paper.

Author reply: We have remove the section discussing GPR that this comment had originally referred to as a response to Reviewer 1’s suggestion:

Original:

Kernel-based approaches, such as e.g. Gaussian Processes (GP) [1, 2] or Kernel Ridge Regression (KRR) [3], are a way to remedy this problem as they often generalize better from limited sample sizes. However, such methods generally tend to exhibit poor computational scaling with the number of reference configurations, in both training (cubic in training set size) and prediction (linear in training set size). This limits both the amount of training data they can be trained on as well as the length and size of simulations that can be simulated with them.

Reviewer: * Eqns.2-4 are rather obvious and can be condensed.

Author reply: We have condensed the presentation of these well-known equations as follows:

Original:

Given a set of atoms (a molecule or a material), we aim to find a mapping from atomic positions \vec{r}_i and chemical species (identified by atomic numbers Z_i) to the total potential energy and the forces acting on the atoms:

$$f : \{\vec{r}_i, Z_i\} \rightarrow E_{pot}$$

Forces are obtained as gradients of the predicted potential energy with respect to the atomic positions, which guarantees energy conservation:

$$\vec{F}_i = -\nabla_i E_{pot}$$

Gradients can be obtained with relatively low computational overhead in modern auto-differentiation frameworks such as TensorFlow or PyTorch [23, 24].

Following previous work [25], we further define the total potential energy of the system as a sum of atomic potential energies:

$$E_{pot} = \sum_{i \in N_{atoms}} E_{i,atomic}$$

These atomic local energies $E_{i,atomic}$ are the scalar node attributes predicted by the graph neural network.

Changed to:

Given a set of atoms (a molecule or a material), we aim to find a mapping from atomic positions $\{\vec{r}_i\}$ and chemical species $\{Z_i\}$ to the total potential energy E_{pot} and the forces acting on the atoms $\{\vec{F}_i\}$. Following previous work [25], this total potential energy is obtained as a sum of atomic potential energies. Forces are then obtained as the gradients of this predicted total potential energy with respect to the atomic positions (thereby guaranteeing energy conservation):

$$E_{pot} = \sum_{i \in N_{atoms}} E_{i,atomic}$$

$$\vec{F}_i = -\nabla_i E_{pot}$$

The atomic local energies $E_{i,atomic}$ are the scalar node attributes predicted by the graph neural network.

Reviewer: * I am a bit confused by Eqn.5: does it mean that the interacting force vector between two atoms i and j is only dependent on the displacement r_{ij} ? But one would expect that the force also depends on the other atoms close to i and j . Presumably such many-body effects are considered during the convolution step, but this is not clear from Eqn.5.

Author reply: The $F(\vec{r}_{ij})$ in Eqn. 5 refers to the convolution filter, not to the atomic force vector. The final force on atom i is a function of other atoms as well, not only of atom j . We have revised the text to more clearly highlight that this refers to the convolutional filter:

Original:

To achieve rotation equivariance, the convolution filters are constrained to be products of learnable radial functions and spherical harmonics, which are equivariant under SO(3) [26]:

$$F(\vec{r}_{ij}) = R(r_{ij})Y_m^{(l)}(\hat{r}_{ij})$$

where if \vec{r}_{ij} denotes the relative position from central atom i to neighboring atom j , \hat{r}_{ij} and r_{ij} are the associated unit vector and interatomic distance, respectively.

Changed to:

To achieve rotation equivariance, the convolution filters $F(\vec{r}_{ij})$ are constrained to be products of learnable radial functions and spherical harmonics, which are equivariant under $SO(3)$ [26]:

$$F(\vec{r}_{ij}) = R(r_{ij})Y_m^{(l)}(\hat{r}_{ij})$$

where if \vec{r}_{ij} denotes the relative position from central atom i to neighboring atom j , \hat{r}_{ij} and r_{ij} are the associated unit vector and interatomic distance, respectively, and $F(\vec{r}_{ij})$ denotes the corresponding convolutional filter. It should be noted that all learnable weights in the filter lie in the rotationally invariant radial function $R(r_{ij})$.

Reviewer: * Page 9, line 446: The authors wrote "ability to achieve high accuracy on a comparatively small data set opens the door to training models on expensive high-order ab-initio quantum chemical methods." This seems to be a bit overstated, as previous MLPs that trained on CCSD(T) and QMC already exist.

Author reply: We thank the referee for this comment, we have updated the manuscript accordingly and now note that increased sample efficiency makes it easier to develop ML potentials for applications in which high-accuracy reference data may be required:

Original:

Ability to achieve high accuracy on a comparatively small data set opens the door to training models on expensive high-order *ab-initio* quantum chemical methods.

Changed to:

The ability to achieve high accuracy on a comparatively small data set facilitates easier development of Machine Learning Interatomic Potentials on expensive high-order *ab-initio* quantum chemical methods, such as e.g. the coupled cluster method CCSD(T).

Reviewer: * Page 10, line 493: What does it mean to "sampled uniformly"? Does it mean taking a snapshot from every X step of the MD trajectory, or does it mean a random selection, or a uniform sampling in the space of the design matrix?

Author reply: "Sampled uniformly" here simply means randomly sampled from the full reference data set, following a uniform distribution, it does *not* mean sample every X step of the trajectory. We have updated the text to more clearly express this:

Original:

The 133 structures were sampled uniformly from the full data set available online, consisting of water and ice structures, made up of a total of 140,000

frames, coming from the same MD trajectories that were used in the earlier work [27].

Changed to:

The 133 structures were sampled randomly following a uniform distribution from the full data set available online which consists of water and ice structures and is made up of a total of 140,000 frames, coming from the same MD trajectories that were used in the earlier work [27]

Reviewer: * Although it is very impressive that the authors are able to train a MLP for water with 1000x fewer training data compared to DeepMD, bear in mind that the first generation of DeepMD (the un-smooth version) is highly data-inefficient due to the "cusps" introduced by the permutations of atoms. Also the water data set contains the whole MD trajectory, meaning that the training data is highly correlated so significant sparsification can be done before hampering the amount of information contained in the data set. I suggest that the authors consider including these more nuanced points, so that readers who are not familiar with the field won't be misled.

Author reply: We recognize that the authors of the DeepMD work have since proposed a next generation of their work, termed DeepMD-Smooth Edition. Unfortunately, that work does not report results on this water data set and we wanted to show results on such a widely studied system. Regarding the time correlation, we note that this would likely in fact even *help* the DeepMD model, rather than hurt it. Namely, the results reported in their work use 87% of the training data, sampled from MD trajectories, resulting in a test set that likely has strong time correlation with the training set and therefore is not an adequate test of how well the model actually learns the PES. In comparison, NequIP samples 0.087% of the data, therefore can not leverage this time-correlation when evaluated on the test set, and is thus put under a more stringent test. Nevertheless, we agree that this degree of correlation would also imply that it is possible the DeepMD model could successfully be trained on fewer data. We have added the following paragraph to the manuscript in order to more clearly include the raised points to the reader:

Changed to:

We note that the version of DeepMD published in [27] is not smooth, and a smooth version has since been proposed [28]. However, [28] does not report results on the water/ice systems. It would be of interest to investigate the performance of the smooth DeepMD version as a function of training set size.

Reviewer: * Some discussions on how the atomic cutoffs and the number of convolutions affect the results can be very helpful to readers.

Author reply: The original manuscript discusses the important of the choice of radial cutoff.

We have added a comment that we find NequIP to be robust to the choice of number of layers:

Original:

We also found it important to choose the radial cutoff distance r_c appropriately.

Changed to:

We also found it important to choose the radial cutoff distance r_c appropriately for a given system. In addition, we observed the number of layers to not have a strong effect as long as they were set within a reasonable range.

Reviewer: Minor point: * Perhaps better to replot the gor in the left panel of Fig.5 using a xrange of [1,6] Angstroms.

Author reply: We have updated the figure of the $g(r)$ accordingly.

Reviewer 3

Reviewer: This paper report an important advance in the development in the area of machine learning interatomic potentials. In particular, by building an equivariant framework that can act on tensors of different order (scalar, vector, high-order tensors), significant improvements in accuracy for smaller training data sets can be achieved, thus addressing a significant challenge in the construction of ML potentials. As such, I believe this paper will be well-received by the community. I have no major criticisms of this paper and the way it is presented, but I would like to make two suggestions:

Author reply: We thank the reviewer for this very positive review. Both suggestion are addressed in the following on an item-per-item basis.

Reviewer: 1. Presenting tables of MAEs is useful to see the performance of the proposed ML scheme, but I think the reader needs to see some representation of the training data. I realize this might not be easy given the many features captured in the data, but an example could be showing a scatter plot of force magnitudes (since training is done using forces, as suggested in Eq. (9)) or even corresponding energies in the MD-17 data set as a function of some key geometric features [the authors might want to see the recent work of Bogojeski et al. Nature Comm. 11, 5223 (2020) for an example of what could be done]. A few selected molecules could be chosen for the main paper and the rest put into a supplementary information document. This might already be sufficient, although if they can find a convenient and visually intuitive representation of the training data for the extended systems, that would be even better.

Author reply: We thank the reviewer for this insightful comment. Given that the input space of positions and chemical species is high-dimensional, a straightforward visualization of energies or forces in this space is likely to be difficult. In an attempt to visualize the reference data, we have plotted a histogram of force components and potential energies on the revised MD-17 data set in the Appendix C. We have changed the main text accordingly:

Changed to:

Recently, a recomputed version of the original MD-17 data with higher numerical accuracy has been released, termed the revised MD-17 data set [9] (an example histogram of potential energies and force components can be found in Appendix C).

Reviewer: 2. The idea of learnable radial functions in the decomposition of the convolution filters presented in Eq. (5) is an interesting one. Could the authors provide some justification for why it is necessary to learn these using a small NN as opposed to simply expanding them in a basis set or as a polynomial and then learning the coefficients of each as part of the overall training scheme. This

idea has been investigated for learning activation functions in neural networks [see, for example, Goyal, et al. 101, 1-18 (2019)]. Wouldn't this be an easier scheme overall?

Author reply: We thank the reviewer for this interesting comment. It should be noted that the radial neural network $R_c^{(l_f, l_i)}$ learns a mapping from a single scalar, the distance r_{ij} , to the high-dimensional vector of weights for *each* channel in the tensor product indexed by c , l_f , p_f , l_i , and p_i . (This is in contrast to an activation function, which maps a single scalar to a single scalar.) As such, while one could train a polynomial or basis expansion for each output separately, we expect the outputs to be related, and as such a small neural network is well-suited for the task. The scheme suggested by the referee could certainly improve the computational efficiency of this part of the model. In practice, however, we observe that the cost of the radial neural networks is quite small compared to other operations in the model and thus a scheme like this would likely not give significant improvements in speed. Nonetheless it remains an interesting direction to consider for future work.

In addition to the changes outlined above, we have also made a number of additional changes, mostly due to the fact that we had updated the network to be equivariant to the Euclidean group $E(3)$, instead of $SE(3)$ and due to the inclusion of results on energy targets. We have outlined the major changes below:

- We have added symmetry w.r.t. parity to the NequIP model, making it equivariant to the group $E(3)$ instead of $SE(3)$. We have therefore adopted the title as well as a number of discussions of the network architecture:

Original:

SE(3)-Equivariant Graph Neural Networks for Data-Efficient and Accurate Interatomic Potentials

Changed to:

E(3)-Equivariant Graph Neural Networks for Data-Efficient and Accurate Interatomic Potentials

Original:

Even though the output of NequIP is the predicted potential energy E_{pot} , which is invariant under translations and rotations, the network contains *internal features* that are tensors which are equivariant to rotation.

Changed to:

Even though the output of NequIP is the predicted potential energy E_{pot} , which is invariant under translations, reflection, and rotations, the network contains *internal features* that are geometric tensors which are equivariant to rotation and reflection.

Original:

Formally, these features are geometric objects that comprise a direct sum of irreducible representations of the $SO(3)$ symmetry group. Second, the convolutions that operate on these geometric objects are equivariant functions instead of invariant ones, i.e. if a feature at layer k is rotated, then the output of the convolution from layer $k \rightarrow k + 1$ rotates accordingly. In practice, the features are implemented as a dictionary $V_{acm}^{(l)}$ with keys l , where $l = 0, 1, 2, \dots$ is a non-negative integer and is called the “rotation order”, labeling the irreducible representations. The indices a, c, m , correspond to the atoms, the channels (elements of the feature), and the representation index which takes values $m \in [-l, l]$, respectively.

Changed to:

Formally, the feature vectors are geometric objects that comprise a direct sum

of irreducible representations of the $O(3)$ symmetry group. The feature vectors $V_{acm}^{(l,p)}$ are indexed by keys l, p , where the ‘‘rotation order’’ $l = 0, 1, 2, \dots$ is a non-negative integer and parity is one of $p \in (1, -1)$ which together label the irreducible representations of $O(3)$. The indices a, c, m , correspond to the atoms, the channels (elements of the feature vector), and the representation index which takes values $m \in [-l, l]$, respectively. The convolutions that operate on these geometric objects are equivariant functions instead of invariant ones, i.e. if a feature at layer k is transformed under a rotation or parity transformation, then the output of the convolution from layer $k \rightarrow k + 1$ is transformed accordingly.

Original:

Finally, as outlined in [26], a full convolutional layer \mathcal{L} implementing an interaction with filter f acting on an input i producing output o : $l_f \otimes l_i \rightarrow l_o$ is given by:

$$\mathcal{L}_{acm_o}^{(l_o)}(\vec{r}_a, V_{acm_i}^{(l_i)}) = \sum_{m_f, m_i} C_{(l_f, m_f)(l_i, m_i)}^{(l_o, m_o)} \sum_{b \in S} R_c^{(l_f, l_i)}(r_{ab}) Y_{m_f}^{(l_f)}(\hat{r}_{ab}) V_{bcm_i}^{(l_i)}$$

where a and b index the central atom of the convolution and the neighboring atom $b \in S$, respectively, and C indicates the Clebsch-Gordan coefficients. As an example of this, we write out a full $1 \otimes 1 \rightarrow 1$ operation (corresponding to a cross-product) in the Methods section. After every convolution, output tensors of a rotation order l stemming from different tensor products are concatenated on a per-atom basis. To update atomic features, the model also leverages self-interaction layers similar to SchNet [6], corresponding to dense layers that are applied in an atom-wise fashion with weights shared across atoms. While different weights are used for different rotation orders, the same set of weights is applied for all representation indices m of a given rotation order l . Shifted softplus nonlinearities [6] are used as rotation-equivariant nonlinearities as introduced in [26], which are applied to the Euclidean norm of the input feature, the output of which is in turn combined with the input tensor, thus preserving overall equivariance.

Changed to:

The final symmetry the network needs to respect is that of parity: how the tensor transformations when the input is mirrored, i.e. $\vec{x} \rightarrow -\vec{x}$. A tensor has even parity ($p = 1$) if it is invariant to such a transformation; it has odd parity ($p = -1$) if its sign flips under that transformation. Parity equivariance is achieved by only allowing contributions from a filter and an incoming tensor feature with parities p_f and p_i to contribute to an output feature if the following selection rule is satisfied:

$$p_o = p_i p_f$$

Finally, as outlined in [26], a full convolutional layer \mathcal{L} implementing an interaction with filter f acting on an input i producing output o : $l_i \otimes l_f \rightarrow l_o$ is

given by:

$$\mathcal{L}_{acm_o}^{(l_o, p_o, l_f, l_i, p_f, p_i)}(\vec{r}_a, V_{acm_i}^{(l_i, p_i)}) = \sum_{m_f, m_i} C_{(l_i, m_i)(l_f, m_f)}^{(l_o, m_o)} \sum_{b \in S} \left(R_c^{(l_f, l_i, p_f, p_i)}(r_{ab}) \right) Y_{m_f}^{(l_f)}(\hat{r}_{ab}) V_{bcm_i}^{(l_i, p_i)}$$

where a and b index the central atom of the convolution and the neighboring atom $b \in S$, respectively, and C indicates the Clebsch-Gordan coefficients. Note that the Clebsch-Gordan coefficients do not depend on the parity of the arguments. There can be multiple $\mathcal{L}_{acm_o}^{(l_o, p_o)}$ tensors for a given output rotation order and parity (l_o, p_o) resulting from different combinations of (l_i, p_i) and (l_f, p_f) ; we take all such possible output tensors with $l_o \leq l_{\max}$ and concatenate them. We also divide the output of the sum over neighbors by \sqrt{N} , where N denotes the average number of neighbors of an atom. To update the atomic features, the model also uses dense layers that are applied in an atom-wise fashion with weights shared across atoms, similar to the self-interaction layers in SchNet [6]. While different weights are used for different rotation orders, the same set of weights is applied for all representation indices m of a given tensor with rotation order l to maintain equivariance.

- We have added a series of additional methods that have been proposed after the first version of this work:

Changed to:

After a first version of this manuscript appeared online [29], a series of other equivariant GNN-IPs have been proposed, such as PaiNN [14] and NewtonNet [15]. Both of these methods were proposed after NequIP and only make use of $l = 1$ tensors. In addition, we also compare a series of other works that have since been proposed, including the GemNet [13], SpookyNet [?], and UNiTE approaches [16].

- We have updated the interaction block and changed the text accordingly:

Original:

Interaction Block: interaction blocks encode interactions between neighboring atoms: the core of this block is the convolution function, outlined in equation 8. For every output rotation order l_o , the features from different tensor product interactions are concatenated to give a new feature, which is in return refined with atom-wise self-interaction layers and equivariant non-linearities. We equip interactions blocks with a ResNet-style update [30] where the input feature \mathbf{x} is updated atom-wise via the output of an interaction block $f(\mathbf{x})$ that gives the final feature $r(\mathbf{x}) = f(\mathbf{x}) + \mathbf{x}$ (features are added element-wise in the m -dimension). Note that this operation is equivariant since the addition of an equivariant feature \mathbf{x} and an equivariant function $f(\mathbf{x})$ preserves equivariance.

While later interaction blocks include all five interactions outlined above, the first interaction block operates on the $l = 0$ embedding with a $0 \otimes 0 \rightarrow 0$ and a $0 \otimes 1 \rightarrow 1$ only.

Changed to:

Interaction Block: interaction blocks encode interactions between neighboring atoms: the core of this block is the convolution function, outlined in equation 8. Features from different tensor product interactions that yield the same rotation and parity pair (l_o, p_o) are mixed by linear atom-wise self-interaction layers. We equip interaction blocks with a ResNet-style update [30]: $\mathbf{x}^{k+1} = f(\mathbf{x}^k) + \text{Self-Interaction}(\mathbf{x}^k)$, where f is the series of self-interaction, convolution, concatenation, and self-interaction. The weights of the Self-Interaction are learned separately for each species. Finally, the mixed features are processed by an equivariant SiLU-based gate nonlinearity [31, 32] (even and odd scalars are not gated, but instead are processed on directly by SiLU and tanh nonlinearities, respectively).

- More accurate reflect the abbreviation here:

Original:

Recently, rotationally invariant graph neural networks (GNN-IPs) have

Changed to:

Recently, rotationally invariant graph neural network interatomic potentials (GNN-IPs) have

- We have updated the radial function to be a more general multi-layer perceptron, instead of a neural network with one hidden layer and have adjusted the discussion accordingly:

Original:

This radial function is implemented as a small neural network with one hidden layer and a shifted softplus activation function [6], operating on interatomic distances expressed in a basis of choice, $R(r_{ij}) : R^{N_b} \rightarrow R^h$, where N_b is the number of basis functions and h is the feature dimension: $R(r_{ij}) = W_2 \ln(0.5 \exp(W_1 B(r_{ij})) + 0.5)$ where $W_1 \in R^{N_{hidden} \times N_b}$ and $W_2 \in R^{h \times N_{hidden}}$ are weight matrices, h is the dimension of the feature and N_{hidden} is the dimension of the hidden layer in the feed-forward neural network (in our experiments, we use $N_{hidden} = N_b$, resulting in comparatively small neural networks for the radial function).

Changed to:

This radial function is implemented as a small multi-layer perceptron that operates on interatomic distances expressed in a basis of choice, $R(r_{ij}) : R \rightarrow R^h$, where h is the feature dimension:

$$R(r_{ij}) = W_n \sigma(\dots \sigma(W_2 \sigma(W_1 B(r_{ij}))))$$

where $B(r_{ij})$ is a basis embedding of the interatomic distance, W_i are weight matrices and $\sigma(x)$ denotes the nonlinear activation function, for which we use the SiLU activation function [31] in our experiments.

- We have updated the discussion of our results on the water system and the comparison to DeepMD to reflect the updated results including energies and the different weightings of the loss function:

Original:

Table II shows the comparison of the predictive force accuracy of NequIP trained on the 133 structures vs DeepMD trained on 133,500 structures. We find that with 1000x fewer training data, NequIP significantly outperforms DeepMD on all four parts of the data set.

Changed to:

Table IV compares the energy and force errors of NequIP trained on the 133 structures vs DeepMD trained on 133,500 structures. We find that with 1000x fewer training data NequIP significantly outperforms DeepMD on all four parts of the data set in the error on the force components. We note that there are $3N$ force components for each training frame but only one energy target. Consequently, one would expect that on energies the much larger training set used for DeepMD would result in an even stronger difference. We find that while this is indeed the case, the NequIP results on the liquid phase are surprisingly competitive. Finally, we report results using three different weightings of energies and forces in the loss function and see that increasing the energy weighting results in significantly improved energy errors at the cost of a small increase in force error. We note that the version of DeepMD published in [27] is not smooth, and a smooth version has since been proposed [28]. However, [28] does not report results on the water/ice systems. It would be of interest to investigate the performance of the smooth DeepMD version as a function of training set size.

- We have updated the discussion of hyperparameters in the Methods section:

Original:

Hyperparameters. Training of models was performed on NVIDIA Tesla

V100 GPUs. Throughout all experiments shown in this work, we use a feature dimension of $h = 64$, 6 interaction blocks, $N_b = 8$ Bessel basis functions and radial neural networks with one hidden layer, also of hidden dimension $N_{hidden} = 8$, giving light-weight radial functions with a comparatively small number of parameters. The final interaction block is followed by the output block, which first reduces the feature dimension to 16 through a self-interaction layer. Finally, through another self-interaction layer, the feature dimension is reduced to a single scalar output value associated with each atom which is then summed over to give the total potential energy. Weights were initialized with the uniform Xavier initialization in the radial networks and orthogonal initialization in the self-interaction layers, biases were initialized with a constant value of 0. In all experiments, we use the Adam optimizer [33] with the TensorFlow 1.14 default settings of $\beta_1 = 0.9$, $\beta_2 = 0.999$, and $\epsilon = 10^{-8}$. We decrease the initial learning rate of 0.001 by a decay factor of 0.8 whenever the validation RMSE in the forces has not seen an improvement for a given number of epochs: for the small molecule tasks, we set this learning rate patience to 1,000, for all other tasks we use 100. We continuously save the model with the best validation RMSE and use the model with the overall best RMSE for evaluation on the test set and MD simulations. We stop the training if either a maximum number of 50,000 epochs (one epochs equals a full pass over the training set) has been reached, or the validation force RMSE has not improved for 2,500 epochs, or the maximum training time has been exceeded, whichever occurs first. All systems were trained for a maximum of 8 days (consisting of four runs of 48-hour time-limited compute jobs, which are restarted from the best saved model, i.e. potentially including repeats in the training) with the exception of the $\text{Li}_4\text{P}_2\text{O}_7$, which was trained for 12 days (six 48-hour compute jobs) and the LiPS systems, which were trained for 4 days (two 48-hour compute jobs). We use a batch size of 5 structures for all small molecule tasks, and a batch size of 1 structure for all other tasks. We found small batch sizes to be important for obtaining high predictive accuracy. We also found it important to choose the radial cutoff distance r_c appropriately. A list of the cutoff radii in units of [\AA] that were used for the different systems is given in Table VII.

Changed to:

Hyperparameters. All models were trained on a NVIDIA Tesla V100 GPU in single-GPU training. For the small molecule systems, we use 5 interaction blocks, a learning rate of 0.01 and a batch size of 5. For the periodic systems, we use 6 interaction blocks, a learning rate of 0.005 and a batch size of 1. We decrease the initial learning rate by a decay factor of 0.8 whenever the validation loss in the forces has not seen an improvement for 50 epochs. We continuously save the model with the best validation loss in the forces and use the model with the overall best validation loss for evaluation on the test set and MD simulations. For validation and test error evaluation, we use an exponential moving average of the training weights with weight 0.99. Training is stopped

if either of the following conditions is met: a) a maximum training time of 7 days is reached; b) a maximum number 1,000,000 epochs is reached; c) the learning rate drops below 10^{-6} ; d) the validation loss does not improve for 1,000 epochs. We note that competitive results can typically be obtained within a matter of hours or often even minutes and most of the remaining training time is spent on only small improvements in the errors. We found the use of small batch sizes to be an important hyperparameter. We also found it important to choose the radial cutoff distance r_c appropriately for a given system. In addition, we observed the number of layers to not have a strong effect as long as they were set within a reasonable range. We use different numbers of l and feature dimensions for different systems and similarly also vary the cutoff radius for different systems. A full outline of the choices for l , feature size, cutoff radius as well as the weights for energies and forces in the loss function can be found in VII. All models were trained with both even and odd features. The weights were initialized according to a standard normal distribution (for details, see the `e3nn` software implementation [34]). The invariant radial networks act on a trainable Bessel basis of size 8 and were implemented with 3 hidden layers of 64 neurons with SiLU nonlinearities between them. The even scalars of the final interaction block are passed to the output block, which first reduces the feature dimension to 16 even scalars through a self-interaction layer. Finally, through another self-interaction layer, the feature dimension is reduced to a single scalar output value associated with each atom which is then summed over to give the total potential energy. Forces are obtained as the negative gradient of this predicted total potential energy, computed via automatic differentiation. All models were optimized with Adam with the AMSGrad variant in the PyTorch implementation [33, 35, 36] with $\beta_1 = 0.9$, $\beta_2 = 0.999$, and $\epsilon = 10^{-8}$ without weight decay. The average number of neighbors used for the $\frac{1}{\sqrt{N}}$ normalization of the convolution was computed over the full training set. For all molecular results, the average number of neighbors was computed once on the $N=1000$ case for revised MD-17 and used for all other experiments. For the water sample efficiency and the LiPS experiments it was computed once on the $N=1000$ and $N=2500$ cases, respectively and then used for all other experiments for that system. All input files for training of NequIP models will be shared upon publication.

- The computational efficiency of the method is a function of different choices of hyperparameters as well as the current software implementation, which we are consistently improving. This warrants a separate, more in-depth study on the trade-offs between accuracy and computational efficiency. In addition, we have rewritten the entire code base for NequIP and have changed from a backend based on TensorFlow to one based on PyTorch and the `e3nn` library, which in turn results in a changed computational efficiency. We have therefore removed the section *Computational Efficiency* from the manuscript, together with table VI that reported the corresponding timings:

Original:

Finally, we report the computational efficiency of NequIP and compare it to that of the *ab-initio* methods on two examples shown in this work: for a molecular system, we choose the Toluene molecule, computed at the CCSD(T)-level of theory [5]; for a material with periodic boundary conditions, we choose the Formate on Cu system, in which reference data were obtained with DFT. For both systems, we report the time required for a single force call on a CPU node with 32 cores. The results are shown in Table VI. In both cases, NequIP gives a large speed-up over the *ab-initio* methods. In the case of the Toluene system, this means that 58.4 minutes of a NequIP simulation can obtain the simulation time equaling one century of a CCSD(T) simulation.

References

- [1] Bartók, A. P., Payne, M. C., Kondor, R. & Csányi, G. Gaussian approximation potentials: The accuracy of quantum mechanics, without the electrons. *Physical review letters* **104**, 136403 (2010).
- [2] Vandermause, J. *et al.* On-the-fly active learning of interpretable bayesian force fields for atomistic rare events. *npj Computational Materials* **6**, 1–11 (2020).
- [3] Chmiela, S. *et al.* Machine learning of accurate energy-conserving molecular force fields. *Science advances* **3**, e1603015 (2017).
- [4] Schütt, K. T., Arbabzadah, F., Chmiela, S., Müller, K. R. & Tkatchenko, A. Quantum-chemical insights from deep tensor neural networks. *Nature Communications* **8**, 13890 (2017). URL <https://doi.org/10.1038/ncomms13890>.
- [5] Chmiela, S., Sauceda, H. E., Müller, K.-R. & Tkatchenko, A. Towards exact molecular dynamics simulations with machine-learned force fields. *Nature Communications* **9**, 3887 (2018).
- [6] Schütt, K. *et al.* Schnet: A continuous-filter convolutional neural network for modeling quantum interactions. In *Advances in neural information processing systems*, 991–1001 (2017).
- [7] Klicpera, J., Groß, J. & Günnemann, S. Directional message passing for molecular graphs. *arXiv preprint arXiv:2003.03123* (2020).
- [8] Christensen, A. S., Bratholm, L. A., Faber, F. A. & Anatole von Lilienfeld, O. Fchl revisited: Faster and more accurate quantum machine learning. *The Journal of Chemical Physics* **152**, 044107 (2020).
- [9] Christensen, A. S. & von Lilienfeld, O. A. On the role of gradients for machine learning of molecular energies and forces. *Machine Learning: Science and Technology* **1**, 045018 (2020).
- [10] Devereux, C. *et al.* Extending the applicability of the ani deep learning molecular potential to sulfur and halogens. *Journal of Chemical Theory and Computation* **16**, 4192–4202 (2020).
- [11] Drautz, R. Atomic cluster expansion for accurate and transferable interatomic potentials. *Physical Review B* **99**, 014104 (2019).
- [12] Unke, O. T. *et al.* Spookynet: Learning force fields with electronic degrees of freedom and nonlocal effects. *Nature Communications* **12**, 7273 (2021). URL <https://doi.org/10.1038/s41467-021-27504-0>.
- [13] Klicpera, J., Becker, F. & Günnemann, S. Gemnet: Universal directional graph neural networks for molecules. *arXiv preprint arXiv:2106.08903* (2021).

- [14] Schütt, K. T., Unke, O. T. & Gastegger, M. Equivariant message passing for the prediction of tensorial properties and molecular spectra. *arXiv preprint arXiv:2102.03150* (2021).
- [15] Haghightalari, M. *et al.* Newtonnet: A newtonian message passing network for deep learning of interatomic potentials and forces. *arXiv preprint arXiv:2108.02913* (2021).
- [16] Qiao, Z. *et al.* Unite: Unitary n-body tensor equivariant network with applications to quantum chemistry. *arXiv preprint arXiv:2105.14655* (2021).
- [17] Kovács, D. P. *et al.* Linear atomic cluster expansion force fields for organic molecules: beyond rmse. *Journal of Chemical Theory and Computation* (2021).
- [18] Mo, Y., Ong, S. P. & Ceder, G. First principles study of the li10gep2s12 lithium super ionic conductor material. *Chemistry of Materials* **24**, 15–17 (2012).
- [19] Chu, I.-H. *et al.* Insights into the performance limits of the li7p3s11 superionic conductor: a combined first-principles and experimental study. *ACS applied materials & interfaces* **8**, 7843–7853 (2016).
- [20] Cheng, B., Engel, E. A., Behler, J., Dellago, C. & Ceriotti, M. Ab initio thermodynamics of liquid and solid water. *Proceedings of the National Academy of Sciences* **116**, 1110–1115 (2019).
- [21] Hestness, J. *et al.* Deep learning scaling is predictable, empirically. *arXiv preprint arXiv:1712.00409* (2017).
- [22] Unke, O. T. & Meuwly, M. Physnet: A neural network for predicting energies, forces, dipole moments, and partial charges. *Journal of chemical theory and computation* **15**, 3678–3693 (2019).
- [23] Abadi, M. *et al.* Tensorflow: Large-scale machine learning on heterogeneous distributed systems. *arXiv preprint arXiv:1603.04467* (2016).
- [24] Paszke, A. *et al.* Pytorch: An imperative style, high-performance deep learning library. In *Advances in neural information processing systems*, 8026–8037 (2019).
- [25] Behler, J. & Parrinello, M. Generalized neural-network representation of high-dimensional potential-energy surfaces. *Physical review letters* **98**, 146401 (2007).
- [26] Thomas, N. *et al.* Tensor field networks: Rotation-and translation-equivariant neural networks for 3d point clouds. *arXiv preprint arXiv:1802.08219* (2018).
- [27] Zhang, L., Han, J., Wang, H., Car, R. & Weinan, E. Deep potential molecular dynamics: a scalable model with the accuracy of quantum mechanics. *Physical review letters* **120**, 143001 (2018).

- [28] Zhang, L. *et al.* End-to-end symmetry preserving inter-atomic potential energy model for finite and extended systems. *Advances in Neural Information Processing Systems* **31** (2018).
- [29] Batzner, S. *et al.* Se(3)-equivariant graph neural networks for data-efficient and accurate interatomic potentials. *arXiv preprint arXiv:2101.03164v1* (2021).
- [30] He, K., Zhang, X., Ren, S. & Sun, J. Deep residual learning for image recognition. In *Proceedings of the IEEE conference on computer vision and pattern recognition*, 770–778 (2016).
- [31] Hendrycks, D. & Gimpel, K. Gaussian error linear units (gelus). *arXiv preprint arXiv:1606.08415* (2016).
- [32] Weiler, M., Geiger, M., Welling, M., Boomsma, W. & Cohen, T. S. 3d steerable cnns: Learning rotationally equivariant features in volumetric data. In *Advances in Neural Information Processing Systems*, 10381–10392 (2018).
- [33] Kingma, D. P. & Ba, J. Adam: A method for stochastic optimization. *arXiv preprint arXiv:1412.6980* (2014).
- [34] Geiger, M. *et al.* e3nn/e3nn: 2021-05-04 (2021). URL <https://doi.org/10.5281/zenodo.4735637>.
- [35] Loshchilov, I. & Hutter, F. Decoupled weight decay regularization. *arXiv preprint arXiv:1711.05101* (2017).
- [36] Reddi, S. J., Kale, S. & Kumar, S. On the convergence of adam and beyond. *arXiv preprint arXiv:1904.09237* (2019).

REVIEWERS' COMMENTS

Reviewer #1 (Remarks to the Author):

After reading the authors' replies to all the comments and the changes made to the text and plots, I consider that the manuscript has definitely improved. My only concern is the fact that, still, the discussion section is weak compared to what is expected to appeal to the broad audience of Nat. Commun. In the revised version, this section only got minor changes. In my opinion, here the authors should discuss several things, such as the context in which such methodological contribution is in, and its role to move the field forward, its interpretability, advantages/disadvantages against models that include explicitly analytical interactions, is there a difference when dealing/describing bonded and non-bonded interactions, etc. Additionally, topics such as possible directions to further develop the method as well as challenges to be addressed should be discussed in this section.

In its current state, I think the article is missing such discussion to complement the numeric results and to have a round and solid manuscript.

Reviewer #3 (Remarks to the Author):

I have read the revised manuscript and responses to the reviews, and I am satisfied with the current version. I recommend publication of the manuscript at this point.

E(3)-Equivariant Graph Neural Networks for Data-Efficient and Accurate Interatomic Potentials

Simon Batzner[†], Albert Musaelian[†], Lixin Sun[†], Mario Geiger^{¶,‡‡}, Jonathan P. Mailoa[‡], Mordechai Kornbluth[‡], Nicola Molinari[†], Tess Smidt^{*,§}, and Boris Kozinsky^{†,‡}

[†] John A. Paulson School of Engineering and Applied Sciences, Harvard University, Cambridge, MA 02138, USA

[¶] Research Laboratory of Electronics, Massachusetts Institute of Technology, Cambridge, MA 02139, USA

^{‡‡} École Polytechnique Fédérale de Lausanne, 1015 Lausanne, Switzerland

^{*} Computational Research Division and Center for Advanced Mathematics for Energy Research Applications, Lawrence Berkeley National Laboratory, Berkeley, CA 94720, USA

[§] Department of Electrical Engineering and Computer Science and Research Laboratory of Electronics, Massachusetts Institute of Technology, Cambridge, MA 02139, USA

[‡] Robert Bosch Research and Technology Center, Cambridge, MA 02139, USA

We thank the reviewers for their feedback and are happy to see that a suitably revised version of the manuscript is in principle accepted for publication. As asked for in the Author Checklist, we provide below a brief summary of the main findings, Twitter handles, and hashtags:

Summary:

An E(3)-equivariant Deep Learning Interatomic Potential is introduced for accelerating Molecular Simulations. The method obtains state-of-the-art accuracy, can faithfully describe dynamics of complex systems, and shows remarkable sample efficiency.

Twitter handles: @simonbatzner, @mario1geiger, @tesssmidt, @bkoz37, @Materials_Intel

Hashtags: #compchem #GNN

In response to reviewer 1's request, we have outlined the changes below:

Reviewer 1

Reviewer: After reading the authors' replies to all the comments and the changes made to the text and plots, I consider that the manuscript has definitely improved. My only concern is the fact that, still, the discussion section is weak compared to what is expected to appeal to the broad audience of Nat. Commun. In the revised version, this section only got minor changes. In my opinion, here the authors should discuss several things, such as the context in which such methodological contribution is in, and its role to move the field forward, its interpretability, advantages/disadvantages against models that include explicitly analytical interactions, is there a difference when dealing/describing bonded and non-bonded interactions, etc. Additionally, topics such as possible directions to further develop the method as well as challenges to be addressed should be discussed in this section.

In its current state, I think the article is missing such discussion to complement the numeric results and to have a round and solid manuscript.

Author reply: We thank the reviewer for this positive review and their comment. We have revised the manuscript as follows:

- We have included a discussion of bonded and non-bonded interactions in the Introduction section:

Original:

The construction of flexible models of the interatomic potential energy based on Machine Learning (ML-IP), and in particular Neural Networks (NN-IP), has shown great promise in providing a way to move past this dilemma, promising to learn high-fidelity potentials from *ab-initio* reference calculations while retaining favorable computational efficiency [1, 2, 3, 4, 5, 6, 7, 8, 9, 10]. One of the limiting factors of NN-IPs is that they typically require collection of large training sets of *ab-initio* calculations, often including thousands or even millions of reference structures [1, 11, 6, 12, 7, 13]. This computationally expensive process of training data collection has severely limited the adoption of NN-IPs as it quickly becomes a bottleneck in the development of force-fields for new systems.

Changed to:

The construction of flexible models of the interatomic potential energy based on machine learning, and in particular neural networks, has shown great promise in providing a way to move past this dilemma, promising to learn high-fidelity potentials from *ab-initio* reference calculations while retaining favorable computational efficiency [1, 2, 3, 4, 5, 6, 7, 8, 9, 10]. Another central difference

to classical force-fields based on analytical functions is that they often consist of explicit bonded and non-bonded terms, whereas machine learning interatomic potentials (ML-IPs) are agnostic to the bond topology of the system and treat all interactions in an identical manner, based on relative interatomic positions and the interacting chemical species. One of the limiting factors of neural network interatomic potentials (NN-IPs) is that they typically require large training sets of *ab-initio* calculations, often including thousands or even millions of reference structures [1, 11, 6, 12, 7, 13]. This computationally expensive process of training data collection has severely limited the adoption of NN-IPs, as it quickly becomes a bottleneck in the development of force-fields for complex systems.

- We have updated the Discussion section to discuss the context of our methodological contribution, interpretability, advantages/disadvantages in comparison to analytical models and possible directions for future work:

Original:

We demonstrate that the Neural Equivariant Interatomic Potential (NequIP), a new type of graph neural network built on E(3)-equivariant convolutions, exhibits state-of-the-art accuracy and exceptional data efficiency on data sets of small molecules and periodic materials. We isolate that the improvements are due to the introduction of equivariant representations in place of more widely used invariant representations. This raises questions about the optimal way to include symmetry in Machine Learning for molecules and materials. A better understanding of why equivariance enables improved sample efficiency and accuracy is likely to be a fruitful direction towards designing better ML algorithms for the construction of Potential Energy Surfaces. In addition to open questions around the effect of equivariance on accuracy and learning dynamics, a clear theoretical understanding of how the many-body character of interactions arises in Message Passing Interatomic Potentials remains elusive. We expect the proposed method will enable researchers in computational chemistry, physics, biology, and materials science to conduct molecular dynamics simulations of complex reactions and phase transformations at increased accuracy and efficiency.

Changed to:

This work introduces NequIP, a novel Machine Learning method for computing the potential energy and atomic forces of molecules and materials based on E(3)-Equivariant Neural Networks. The findings lead to a series of interesting questions to consider: of particular interest is the sample efficiency of the equivariant NequIP network when compared to the more widely used invariant representations. In addition to questions around the effect of equivariance on accuracy and learning dynamics, a clear theoretical understanding of how the many-body character of interactions arises in message passing interatomic potentials remains elusive. Further, a promising direction for future work is

to investigate the potential benefits of explicitly including long-range interactions and to measure to what extent - if any - these might be captured by the message passing mechanism. Finally, while we find that NequIP displays excellent predictive accuracy, generalization to unseen phases, and remarkably high sample efficiency, an open challenge that remains is the interpretability of deep learning interatomic potentials. Energy contributions in classical interatomic potentials can be explicitly assigned to individual types of interactions, such as pair-wise bonded terms or Coulomb or van der Waals non-bonded interactions. The potential benefits and optimal ways of including such physical knowledge explicitly into the complex functional forms underlying deep learning interatomic potentials still need to be systematically explored. On the other hand, the simplicity of the functional form of classical force-fields that allows for this level of interpretability severely limits their accuracy, presenting an interesting tension between the two approaches. We expect the proposed method will enable researchers in computational chemistry, physics, biology, and materials science to conduct molecular dynamics simulations of complex reactions and phase transformations at increased accuracy and efficiency.

Reviewer 3

I have read the revised manuscript and responses to the reviews, and I am satisfied with the current version. I recommend publication of the manuscript at this point.

We thank the reviewer for this positive review.

We have additionally also changed the language and notation in a few places to be more explicit and have revised tables I-IV, table VI and figures 7, 8, 10, and 11 to be in line with a more conventional scheme of computing the errors. Larger changes are outlined below:

- In response to editorial request, we have removed Table V and have instead noted the errors in the test:

Original:

The MAE of the predicted forces using a NequIP model trained on 2,500 structures is shown in Table V, demonstrating that NequIP is able to accurately model the interatomic forces for this complex reactive system. A more detailed analysis of the resulting dynamics will be the subject of a separate study.

Changed to:

A NequIP model trained on 2,500 structures obtains MAEs in the force components of 19.9 meV/Å, 71.3 meV/Å, 13.0 meV/Å, and 47.6 meV/Å, on the four elements C, O, H, and Cu, respectively. We find from this an average force MAE of 38.4 meV/Å, equally weighted over these four per-species MAEs, as well as an energy MAE of 0.50 meV/atom, demonstrating that NequIP is able

to accurately model the interatomic forces for this complex reactive system. A more detailed analysis of the resulting dynamics will be the subject of a separate study.

- **Original:**

Training is stopped if either of the following conditions is met: a) a maximum training time of 7 days is reached;

Changed to:

Training is stopped if either of the following conditions is met: a) a maximum training time of approximately 7 days is reached;

- **Original:**

$$\mathcal{L}_{acm_o}^{(l_o, p_o, l_f, p_f, l_i, p_i)}(\vec{r}_a, V_{acm_i}^{(l_i, p_i)}) = \sum_{m_f, m_i} C_{(l_i, m_i)(l_f, m_f)}^{(l_o, m_o)} \sum_{b \in S} \left(R_c^{(l_f, l_i, p_f, p_i)}(r_{ab}) \right) Y_{m_f}^{(l_f)}(\hat{r}_{ab}) V_{bcm_i}^{(l_i, p_i)}$$

Changed to:

$$\mathcal{L}_{acm_o}^{l_o, p_o, l_f, p_f, l_i, p_i}(\vec{r}_a, V_{acm_i}^{l_i, p_i}) = \sum_{m_f, m_i} C_{l_i, m_i, l_f, m_f}^{l_o, m_o} \sum_{b \in S} \left(R(r_{ab})_{c, l_o, p_o, l_f, p_f, l_i, p_i} \right) Y_{m_f}^{l_f}(\hat{r}_{ab}) V_{bcm_i}^{l_i, p_i}$$

- **Original:**

The final symmetry the network needs to respect is that of parity: how the tensor transformations when the input is mirrored, i.e. $\vec{x} \rightarrow -\vec{x}$. A tensor has even parity ($p = 1$) if it is invariant to such a transformation; it has odd parity ($p = -1$) if its sign flips under that transformation.

Changed to:

The final symmetry the network needs to respect is that of parity: how the tensor transforms under inversion, i.e. $\vec{x} \rightarrow -\vec{x}$. A tensor has even parity ($p = 1$) if it is invariant to such a transformation; it has odd parity ($p = -1$) if its sign flips under that transformation.

- We have edited Figure 4 to only plot starting from $\Delta\tau = 1$ ps since the (0,0) point is undefined in log-log space.
- We have made all input files for our models public.

Original:

All input files for training of NequIP models will be shared upon publication.

Changed to:

The input files for training of NequIP models can be found at <https://github.com/mir-group/nequip-input-files>.

References

- [1] Behler, J. & Parrinello, M. Generalized neural-network representation of high-dimensional potential-energy surfaces. *Physical review letters* **98**, 146401 (2007).
- [2] Bartók, A. P., Payne, M. C., Kondor, R. & Csányi, G. Gaussian approximation potentials: The accuracy of quantum mechanics, without the electrons. *Physical review letters* **104**, 136403 (2010).
- [3] Shapeev, A. V. Moment tensor potentials: A class of systematically improvable interatomic potentials. *Multiscale Modeling & Simulation* **14**, 1153–1173 (2016).
- [4] Thompson, A. P., Swiler, L. P., Trott, C. R., Foiles, S. M. & Tucker, G. J. Spectral neighbor analysis method for automated generation of quantum-accurate interatomic potentials. *Journal of Computational Physics* **285**, 316–330 (2015).
- [5] Vandermause, J. *et al.* On-the-fly active learning of interpretable bayesian force fields for atomistic rare events. *npj Computational Materials* **6**, 1–11 (2020).
- [6] Schütt, K. *et al.* Schnet: A continuous-filter convolutional neural network for modeling quantum interactions. In *Advances in neural information processing systems*, 991–1001 (2017).
- [7] Unke, O. T. & Meuwly, M. Physnet: A neural network for predicting energies, forces, dipole moments, and partial charges. *Journal of chemical theory and computation* **15**, 3678–3693 (2019).
- [8] Klicpera, J., Groß, J. & Günnemann, S. Directional message passing for molecular graphs. *arXiv preprint arXiv:2003.03123* (2020).
- [9] Mailoa, J. P. *et al.* A fast neural network approach for direct covariant forces prediction in complex multi-element extended systems. *Nature machine intelligence* **1**, 471–479 (2019).
- [10] Park, C. W. *et al.* Accurate and scalable multi-element graph neural network force field and molecular dynamics with direct force architecture. *arXiv preprint arXiv:2007.14444* (2020).
- [11] Artrith, N. & Kolpak, A. M. Understanding the composition and activity of electrocatalytic nanoalloys in aqueous solvents: A combination of dft and accurate neural network potentials. *Nano letters* **14**, 2670–2676 (2014).
- [12] Zhang, L., Han, J., Wang, H., Car, R. & Weinan, E. Deep potential molecular dynamics: a scalable model with the accuracy of quantum mechanics. *Physical review letters* **120**, 143001 (2018).
- [13] Smith, J. S., Isayev, O. & Roitberg, A. E. Ani-1: an extensible neural network potential with dft accuracy at force field computational cost. *Chemical science* **8**, 3192–3203 (2017).